# Sodium taurocholate cotransporting polypeptide is a functional receptor for human hepatitis B and D virus

Huan Yan[1,2†], Guocai Zhong[2†], Guangwei Xu[2], Wenhui He[2,3], Zhiyi Jing[2], Zhenchao Gao[1,2], Yi Huang[2,3], Yonghe Qi[2], Bo Peng[2], Haimin Wang[2], Liran Fu[2,3], Mei Song[2,3], Pan Chen[2,3], Wenqing Gao[2], Bijie Ren[2], Yinyan Sun[2], Tao Cai[2], Xiaofeng Feng[2], Jianhua Sui[2], Wenhui Li[2]*

[1]Graduate program in School of Life Sciences, Peking University, Beijing, China; [2]National Institute of Biological Sciences, Beijing, China; [3]Graduate program in Chinese Academy of Medical Sciences and Peking Union Medical College, Beijing, China

**Abstract** Human hepatitis B virus (HBV) infection and HBV-related diseases remain a major public health problem. Individuals coinfected with its satellite hepatitis D virus (HDV) have more severe disease. Cellular entry of both viruses is mediated by HBV envelope proteins. The pre-S1 domain of the large envelope protein is a key determinant for receptor(s) binding. However, the identity of the receptor(s) is unknown. Here, by using near zero distance photo-cross-linking and tandem affinity purification, we revealed that the receptor-binding region of pre-S1 specifically interacts with sodium taurocholate cotransporting polypeptide (NTCP), a multiple transmembrane transporter predominantly expressed in the liver. Silencing NTCP inhibited HBV and HDV infection, while exogenous NTCP expression rendered nonsusceptible hepatocarcinoma cells susceptible to these viral infections. Moreover, replacing amino acids 157–165 of nonfunctional monkey NTCP with the human counterpart conferred its ability in supporting both viral infections. Our results demonstrate that NTCP is a functional receptor for HBV and HDV.

*For correspondence: liwenhui@nibs.ac.cn

†These authors contributed equally to this work

Competing interests: The authors have declared that no competing interests exist

## Introduction

Approximately 2 billion people have been infected with human hepatitis B virus (HBV) worldwide. Over 350 million people currently are chronically infected and are at high risk for progression to cirrhosis, liver failure, or cancer. More than 50% of liver cancers worldwide are attributed to HBV infection. HBV-related liver diseases remain a major public health problem, causing approximately 1 million deaths per year. Individuals coinfected with HBV and HDV are at greater risk for rapid progression and severe disease (*Lavanchy, 2004*; *Hughes et al., 2011*). Despite its enormous medical and social relevance, progress in HBV research has been impeded by the lack of understanding of HBV entry by which the virus specifically infects human liver cells. HBV is an enveloped virus containing a small genome of 3.2 kb of partially double-stranded DNA encoding four overlapping reading frames. The HBV envelope consists of the small (S), middle (M), and large (L) envelope proteins, which are multiple transmembrane spanners sharing the same C-terminal domain corresponding to the S protein but differing at their N-terminal domains (*Figure 1A*) (*Heermann et al., 1984*; *Seeger et al., 2007*). HDV is a small satellite RNA virus of HBV carrying all three HBV envelope proteins and can only propagate when coexisting with HBV. The mechanism of viral entry of HDV is believed to be similar to that of HBV, and HDV has been used as a surrogate for studying HBV infection at the entry level (*Barrera et al., 2004*; *Sureau, 2006*; *Hughes et al., 2011*). The L protein and integrity of S protein are critical for HBV

**eLife digest** Liver diseases related to the human hepatitis B virus (HBV) kill about 1 million people every year, and more than 350 million people around the world are infected with the virus. Some 15 million of these people are also infected with the hepatitis D virus (HDV), which is a satellite virus of HBV, and this places them at an even higher risk of liver diseases, including cancer. The viruses are known to enter liver cells by binding to receptors on their surface before being engulfed.

Both HBV and HDV have outer coats that consist of three kinds of envelope proteins, and a region called the pre-S1 domain in one of them is known to have a central role in the interaction between the viruses and the receptors and, therefore, in infecting the cells. However, the identity of the HBV receptor has remained a mystery. Now Yan et al. have identified this receptor to be sodium taurocholate cotransporting polypeptide. This protein, known as NTCP for short, is normally involved in the circulation of bile acids in the body.

In addition to humans, only two species are known to be susceptible to infection by human HBV and HDV—chimpanzees and a small mammal known as the treeshrew. Yan et al. started by isolating primary liver cells from treeshrews, and then used a combination of advanced purification and mass spectrometry analysis to show that the NTCP on the surface of the cells interacts with the pre-S1 domain in HBV.

The authors then performed a series of gene knockdown experiments on liver cells of both human and treeshrew origin: when the gene that codes for NTCP was silenced, HBV infection was greatly reduced. Moreover, they were able to transfect HepG2 cells—which are widely used in research into liver disease, but are not susceptible to HBV and HDV infection—with NTCP from humans and treeshrews to make them susceptible. Similarly, although monkeys are not susceptible to HBV, replacing just five amino acids in monkey NTCP with their human counterparts was enough to make the monkey NTCP a functional receptor for the viruses.

In the past, basic research into HBV and the development of antiviral therapeutics have both been hindered by the lack of suitable in vitro infection systems and animal models. Now, the work of Yan et al. means that it will be possible to use NTCP-complemented HepG2 cells for challenges as diverse as fundamental studies of basic viral entry/replication mechanisms and large-scale drug screening. It is also possible that HBV and HDV infection might interfere with some of the important physiological functions carried out by NTCP, so the latest work could also be of interest to medical scientists working on other diseases related to these infections.

and HDV infections. The pre-S1 domain of the L protein is a key determinant for entry of both HBV and HDV and is believed to mediate viral interaction with the cellular receptor(s) on hepatocytes (*Gripon et al., 1995*; *Le Seyec et al., 1999*; *Chouteau et al., 2001*; *Blanchet and Sureau, 2007*; *Le Duff et al., 2009*). Although a number of HBV receptor candidates have been reported in the past, none has been confirmed to be functional in supporting viral infection (*Glebe and Urban, 2007*).

An N-terminal myristoylated peptide corresponding to amino acids (aa) 2–48 of the pre-S1 domain of the L protein has been shown to effectively block both HBV and HDV infections of hepatocytes through engaging an unknown cellular component, most likely a viral receptor (*Barrera et al., 2005*; *Glebe et al., 2005*; *Gripon et al., 2005*; *Engelke et al., 2006*; *Schulze et al., 2010*). In the current study, by using a synthetic modified peptide originating from the native aa 2–48 lipopeptide (Myr-47/WT) as a probe and employing a series of biochemical approaches and virological assays, we identified and confirmed that sodium taurocholate cotransporting polypeptide (NTCP), a multiple transmembrane transporter mainly expressed in the liver, interacts specifically with the L proteins of HBV and HDV and functions as a common receptor for both viruses.

## Results

### Photoreactive ligand peptides for identification of interacting protein(s) of pre-S1 domain of L envelope protein

To identify the pre-S1 interacting molecule(s), we employed a photo-cross-linking approach using a synthetic peptide derived from the native pre-S1 peptide with particular residues replaced by

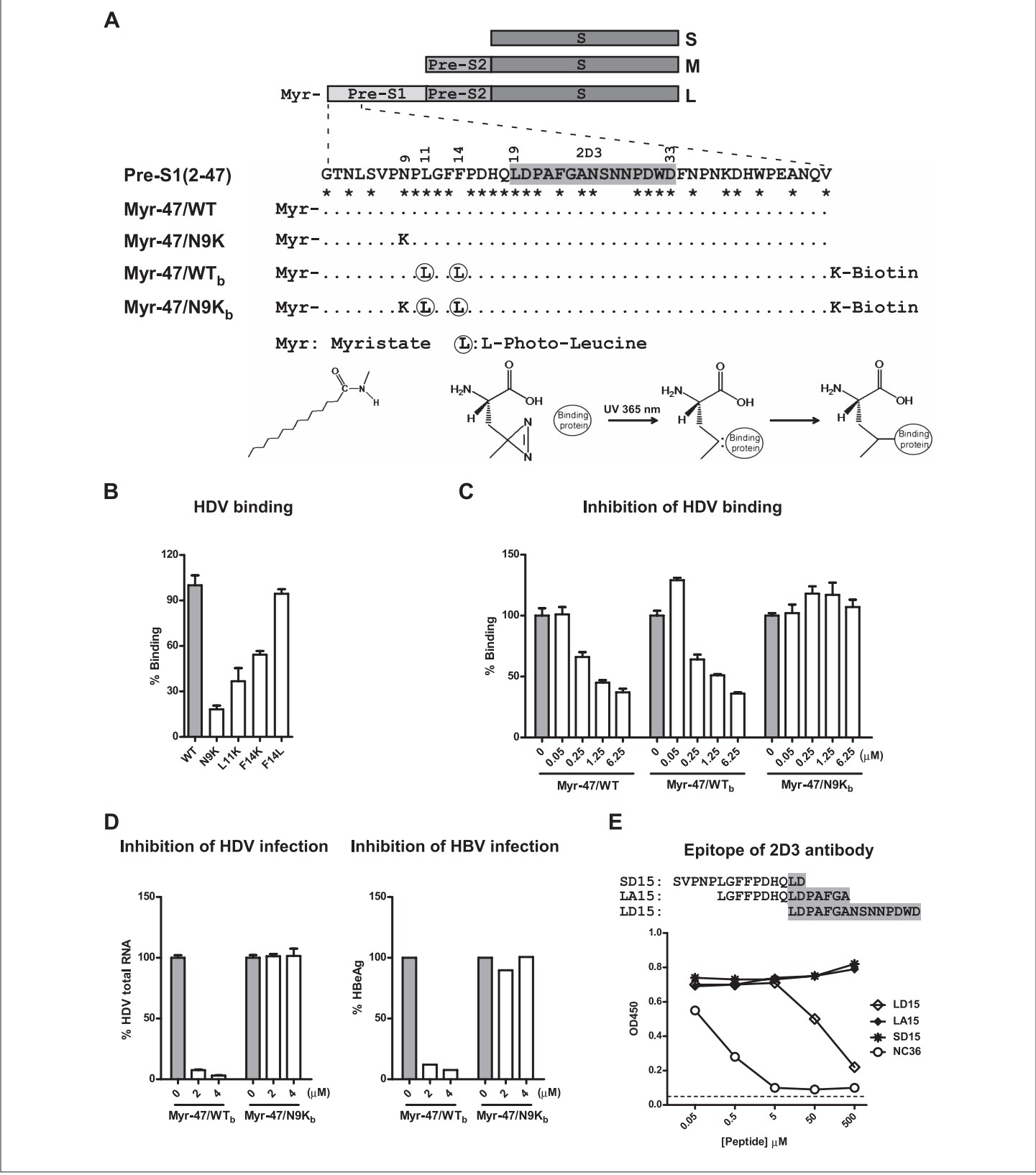

**Figure 1**. Developing photoreactive peptide ligands and an antibody for identifying pre-S1 binding partner(s) by zero distance cross-linking. (**A**) Schematic diagram of HBV envelope proteins and N-terminal peptides of pre-S1 domain. Pre-S1 (2-47): 2-47th residues of the pre-S1 domain of the L

*Figure 1. Continued on next page*

*Figure 1. Continued*

protein of HBV (S472 strain, genotype C). Residue numbering is based on genotype D. Asterisk indicates highly conserved residues among genotypes. Epitope of mAb 2D3 was shaded in gray. (**B**) Effect of alterations of the critical N-terminal residues within pre-S1 region of L protein on HDV binding to PTHs. Both wild-type (WT) and mutant HDV virions carry HBV envelope proteins. Mutant HDV carries point mutation as indicated in the pre-S1 region of L protein. PTHs were incubated with HDV at 16°C for 4 hr and followed by extensive wash; bound virions were quantified by qRT-PCR for virus genome RNA copy, and the data are presented as percentage of virus binding, the binding of WT virus was set as 100%. (**C**) Myr-47/WT$_b$ bait peptide dose-dependently inhibited HDV virion binding. The binding assay was performed similarly as panel B except that PTHs were pre-incubated with indicated peptides. (**D**) Inhibition of viral infection by the photoreactive peptides. Left: PTHs were pre-incubated with peptides at indicated concentrations at 37°C for 1 hr and then inoculated with HDV virus. Viral infection was examined by measuring viral RNA in infected cells with qRT-PCR 6 days post-infection (dpi). Data are presented as percentage HDV infection. Right: peptides at indicated concentrations were added to PTHs before HBV inoculation. The cell culture medium was replenished every 2 days. Secreted viral antigen HBeAg was measured by ELISA on 6 dpi, and the data are presented as percentage of that in the absence of peptides. (**E**) Antibody 2D3 recognizes residues 19–33 of pre-S1. Peptide NC36 (aa 4–36 of pre-S1, NLSVPNPLGFFPDHQLDPAFGANSNNPDWDFNP) conjugated with keyhole limpet hemocyanin (KLH) was the immunogen peptide for generating mouse mAb 2D3. Binding activity of 2D3 with full-length pre-S1 protein was measured by ELISA in the presence of competition peptides at indicated concentrations. LD15 peptide compassing residues 19–33 of pre-S1 inhibited 2D3 binding in a dose-dependent manner, indicating that 2D3 recognizes an epitope within this region. HBV: hepatitis B virus; mAb: monoclonal antibody; HDV: hepatitis D virus; PTH: primary *Tupaia* hepatocytes; HBeAg: HBV e antigen.

nonnatural amino acids (L-photo-leucine, *L-2-amino-4,4-azi-pentanoic acid*) (***Figure 1A***). L-photo-leucine contains a photoactivatable diazirine ring. Irradiation of ultraviolet (UV) light at 365 nm induces a loss of nitrogen of the diazirine ring and yields a reactive carbene group with short half-life for covalent cross-linking at nearly zero distance (***Suchanek et al., 2005***). Primary hepatocytes isolated from treeshrews (*Tupaia belangeri*), the only species susceptible to human HBV infection other than humans and chimpanzees (***Su et al., 1987***; ***Walter et al., 1996***; ***Glebe et al., 2003***), were used as target cells. To maximize the efficiency of photo-cross-linking, two residues (leu$_{11}$ and phe$_{14}$) in a region (aa 9–15) known to be critical for viral infection (***Schulze et al., 2010***) were chosen for substitution with L-photo-leucine. Leu$_{11}$ is 100% conserved among HBV genotypes, and the 14th residue is a phenylalanine in most genotypes but a leucine in some HBV strains of genotypes F and G. Changing phe$_{14}$ to leucine (F14L) did not significantly affect the binding of HDV virion to primary *Tupaia* hepatocytes (PTHs) (***Figure 1B***). The activity of the synthesized peptide ligand Myr-47/WT$_b$ (or WT$_b$ hereafter) containing photo-leucines at positions 11 and 14 was also confirmed (***Figure 1C,D***). WT$_b$ inhibited HDV binding to PTHs with efficiency comparable to Myr-47/WT that is comprised of all natural amino acids (***Figure 1A,C***). A peptide Myr-47/N9K$_b$ (or N9K$_b$ hereafter) similar to WT$_b$ but with an additional mutation at the ninth residue (N9K) did not block HDV binding to PTHs (***Figure 1C***). WT$_b$ but not N9K$_b$ inhibited viral infection of HBV and HDV on PTHs (***Figure 1D***). Both WT$_b$ and N9K$_b$ peptides were myristoylated at the N-terminus and conjugated with a biotin tag on a C-terminal lysine residue (***Figure 1A***). N9K$_b$ differs from WT$_b$ by only one amino acid but completely lost these blocking activities. Thus, N9K$_b$ was used as a negative control for WT$_b$. In addition, a monoclonal antibody (mAb) 2D3, which specifically recognizes an epitope adjacent to the critical receptor-binding region of the peptides and shared by both WT$_b$ and N9K$_b$, was developed (***Figure 1E***).

## Identification of NTCP as a specific binding protein of pre-S1

The WT$_b$ or control N9K$_b$ peptide at 200 nM was then applied to PTHs in culture and near zero distance cross-linking was induced by UV irradiation. The cross-linked peptide and associated partners were precipitated by streptavidin T1 beads and separated by SDS–PAGE. Western blotting using 2D3 as a probe revealed several bands including a major smeared band with apparent molecular weight of ~65 kDa in the WT$_b$ but not N9K$_b$ cross-linked sample. The 65-kDa band shifted to ~43 kDa upon treatment with the deglycosylation enzyme PNGase F (***Figure 2A***, left), indicating that it is highly N-glycosylated. The WT$_b$ cross-linked protein apparently contained no intermolecular disulfide bonds as it migrated similarly under both nonreducing and reducing conditions (***Figure 2A***, right). The non-photoreactive Myr-47/WT peptide but not its N9K mutant peptide effectively competed with WT$_b$ for cross-linking to the 65-kDa band (***Figure 2B***). The cross-linked protein from PTHs decreased in abundance rapidly over time during culture (***Figure 2C***). We also examined primary human hepatocytes (PHHs) in the cross-linking experiments. Bands with slightly smaller molecular weights than those seen in the PTH cells were also observed in PHHs (***Figure 2D***).

We then proceeded to identify the target protein(s) using affinity purification followed by mass spectrometry (MS) analysis. The purification procedure included three tandem steps after photo-cross-linking:

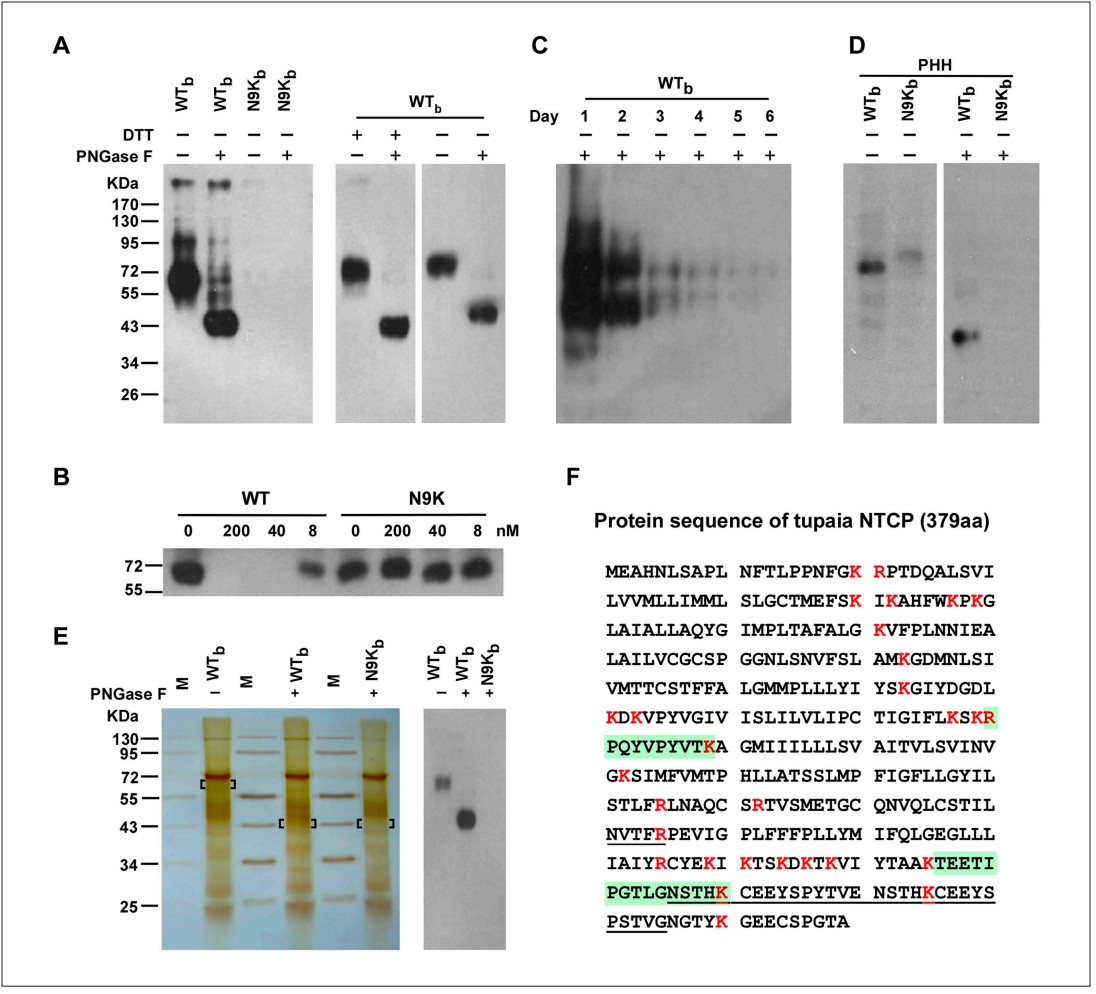

**Figure 2**. Identification of pre-S1 binding protein on primary hepatocytes with photoreactive peptide Myr-47/WT_b. (**A**) Left: Cultured PTHs at 24–48 hr after isolation and plating were photo-cross-linked with 200 nM Myr-47/WT_b (WT_b) or Myr-47/N9K_b (N9K_b), followed by Streptavidin Dynal T1 beads precipitation and Western blot analysis using mAb 2D3. The protein cross-linked by WT_b is sensitive to PNGase F treatment and shifted from ~65 to ~43 kDa. Right: WT_b cross-linked samples were treated with 100 mM DTT and/or PNGase F as indicated and detected similarly as in the left panel. (**B**) Non-photoreactive Myr-47/WT peptide (WT) but not its N9K mutant competed with 200 nM of WT_b peptide for cross-linking with PTHs in a dose-dependent manner. (**C**) The abundance of the target protein(s) in PTH cells decreased over time. PTHs on different days of in vitro culturing were photo-cross-linked with 200 nM WT_b. The cross-linked samples were analyzed by Western blot. The two bands at ~65 and ~43 kDa were due to incomplete deglycosylation by PNGase F. (**D**) WT_b cross-linking with primary human hepatocytes (PHH). Frozen PHH cells were thawed and plated 1 day before cross-linking. With same procedure as in panel A, 200 nM WT_b but not N9K_b cross-linked with a glycoprotein of molecular weight at ~60 kDa, which shifted to ~39 kDa upon PNGase F treatment. (**E**) Purification of target protein(s) for MS analysis. PTHs photo-cross-linked with 200 nM of WT_b or N9K_b peptide were lysed, then the peptides and their cross-linked proteins were purified in tandem with Streptavidin Dynal T1 beads, mAb 2D3 conjugated beads, and Streptavidin Dynal T1 beads in 1× RIPA buffer. Extensive wash was applied for each purification step. The samples were treated with or without PNGase F as indicated prior to the last step of Streptavidin beads precipitation. The final purified samples were subjected to SDS-PAGE followed by silver staining (left). Bracketed areas indicate the bands cut for MS analysis. Western blot analysis (right) of the same cross-linked samples were performed similarly as in panel A. The top 10 nonredundant proteins identified in the 3 samples by MS analysis are listed in *Figure 2—Source data 1*. The common protein hit identified by MS analysis of the ~65- and ~43-kDa bands cut from the WT_b cross-linked sample was *Tupaia* NTCP (tsNTCP), and the representative MS/MS spectra and parameters of the peptide hits are shown in *Figure 2—figure supplement 5*. The control band cut from N9K_b cross-linked sample did not generate any hits on any of these peptides. (**F**) Predicted tsNTCP protein sequence. A 30-amino acid insertion unique to tsNTCP is underlined. Two peptides identified by LC-MS/MS were highlighted in green. All lysine and arginine are highlighted in red to

*Figure 2. Continued on next page*

*Figure 2. Continued*

indicate trypsin cleavage sites. Many of the potential tryptic peptides are not appropriate for LC-MS detection because of unfavorable size and/or hydrophobicity. PTH: primary *Tupaia* hepatocytes; MS: mass spectrometry.

The following source data and figure supplements are available for figure 2.
**Source data 1**. Top 10 nonredundant proteins identified in the three samples (**Figure 2E**) by MS analysis
**Figure supplement 1**. Generation of *Tupaia* hepatocytes proteome database from Illumina deep sequencing-determined transcriptome of PTHs.
**Figure supplement 2**. Generation of *Tupaia* hepatocytes proteome database from Illumina deep sequencing-determined transcriptome of PTHs.
**Figure supplement 3**. Generation of *Tupaia* hepatocytes proteome database from Illumina deep sequencing-determined transcriptome of PTHs.
**Figure supplement 4**. Generation of *Tupaia* hepatocytes proteome database from Illumina deep sequencing-determined transcriptome of PTHs.
**Figure supplement 5**. Representative MS/MS spectra and parameters of the identified peptide hits.

capturing all biotin-labeled proteins with streptavidin T1 beads, sorting out the target protein(s) with 2D3 antibody affinity beads, and then purifying with streptavidin T1 beads again to remove residual molecules that were not covalently cross-linked with the bait peptide. The purified samples were subsequently subjected to SDS-PAGE followed by silver staining. Similar to the Western blotting results with the 2D3 antibody, a ~65-kDa protein band was visible by silver staining. The band was also shifted to ~43 kDa upon PNGase F treatment (**Figure 2E**). Both the original 65-kDa and the shifted 43-kDa bands were subsequently excised from the gel and subjected to LTQ-Orbitrap Velos (Thermo Fisher Scientific, MA. USA) MS analysis after trypsin digestion. The tandem mass spectra were searched against a *Tupaia* hepatocyte protein database, which we had established by deep sequencing of the transcriptome (**Figure 2—figure supplements 1–4**). Two different tryptic peptide fragments, which were identified from both the ~65-kDa and ~43-kDa bands (**Figure 2—figure supplement 5**), matched to a protein homolog of human NTCP. *Tupaia* NTCP (tsNTCP) shares 83.9% protein sequence identity with its human counterpart and has an insertion of 30 aa near its C-terminus (**Figure 2F**). The peptide (TEETIPGTLGNSTH) containing 4 aa of this insertion (underlined) was one of the two peptides identified by the MS analysis at a high confidence level (**Figure 2—figure supplement 5**). These data suggest that NTCP is the protein specifically interacting with the WT$_b$ bait peptide.

## Confirmation of NTCP as a specific binding protein of pre-S1

We next cloned human and *Tupaia NTCPs* and validated the binding of the exogenously expressed NTCPs with the WT$_b$ peptide and an N-terminal myristoylated pre-S1 peptide with native residues. Both human NTCP (hNTCP) and tsNTCP could be efficiently cross-linked by WT$_b$ but not N9K$_b$ when expressed in 293T cells as shown by Western blotting with the anti-WT$_b$ antibody 2D3 as well as an anti-C9 antibody recognizing the C-terminal C9 tag of the recombinant hNTCP and tsNTCP proteins (**Figure 3A**). WT$_b$ but not the control N9K$_b$ peptide bound to 293T cells expressing a green fluorescent protein (GFP)-tagged tsNTCP (tsNTCP-EGFP) and co-localized with tsNTCP-EGFP on the cell surface. This binding was readily competed off by the free Myr-47/WT peptide (**Figure 3B**). Moreover, a native pre-S1 peptide specifically recognized the human hepatocellular carcinoma Huh-7 cell line transfected with hNTCP (**Figure 3C**). Consistently, Huh-7 cells transfected with either tsNTCP or hNTCPs had markedly increased HDV binding to the cells. The Myr-47/WT peptide readily competed with binding of the wild-type HDV, whereas a noninfectious mutant HDV virus bearing a single N9K mutation in the pre-S1 domain of its L envelope protein failed to bind either hNTCP- or tsNTCP-expressing Huh-7 cells (**Figure 3D**). Collectively, these data demonstrated a specific interaction between NTCP and the pre-S1 domain of the L protein, which directly mediates the binding of HDV virions to target cells.

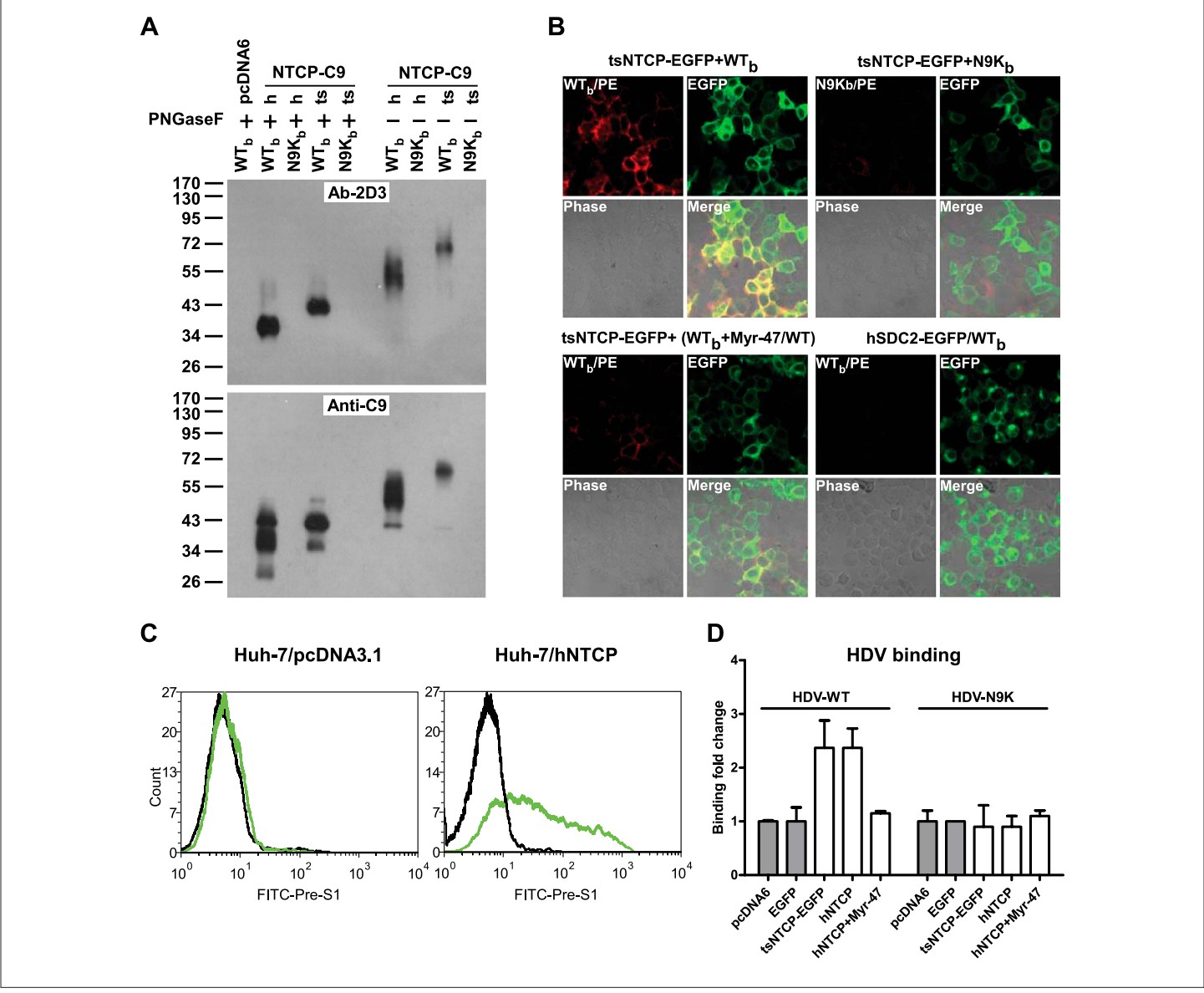

**Figure 3**. Binding of NTCP with N-terminal peptide of pre-S1 and HDV virions. (**A**) 293T cells transfected with an expression vector or plasmid containing cDNA of h*NTCP* or ts*NTCP* fused with a C9 tag at its C-terminus were cross-linked with 200 nM Myr-47/WT_b or Myr-N9K_b similarly as in *Figure 2A* at 24 hr post-transfection. Cross-linked protein samples were precipitated by Streptavidin Dynal beads followed by treatment with PNGase F as indicated, and then analyzed by Western blotting using mAb 2D3 or anti-C9 tag antibody. (**B**) 293T cells transfected with tsNTCP-EGFP or a control hSDC2-EGFP (encoding human heparan sulfate proteoglycan core protein fused with EGFP at C-terminus) expression plasmid were incubated with WT_b or N9K_b in the presence or absence of 200 nM non-photoreactive Myr-47/WT as indicated. Bound peptides were probed with PE-streptavidin and the colocalization of peptide and NTCP on cell surface was shown in the merged images. (**C**) FACS analysis of pre-S1 peptide binding with hNTCP transiently transfected Huh-7 cells. 24 hr post-transfection with hNTCP or a control plasmid, the cells were stained with 200 nM FITC-pre-S1 (FITC-labeled lipopeptide corresponding to the N-terminal 59-amino acid of pre-S1). The binding was analyzed by flow cytometry. (**D**) Huh-7 cells, after 24 hr of transfection of indicated plasmids, were incubated with wild-type HDV or HDV with a N9K mutation on its L protein. Bound virions were quantified by qRT-PCR. The result is presented as fold changes of binding over the background virus binding to pcDNA6-transfected cells. mAb: monoclonal antibody; tsNTCP: *Tupaia* NTCP; NTCP: sodium taurocholate cotransporting polypeptide.

## NTCP expression is required for HBV and HDV infection

To test the requirement of endogenous expression of NTCP for HBV and HDV infection, we first examined the effect of *NTCP* gene silencing on viral infection of PTHs. PTHs were transfected with tsNTCP-specific or a control small interfering RNA (siRNA) prior to viral inoculation. When tsNTCP

mRNA level was reduced to ~30% in tsNTCP siRNA-transfected cells (*Figure 4A*, upper-left), total HDV RNA copies were markedly reduced in these cells comparing to those transfected with control siRNA. We further quantified the HDV genome and antigenome RNA copies using strand-specific reverse transcription followed by quantitative real-time polymerase chain reaction (qPCR). The HDV antigenome is a circular replication intermediate that is complementary to the genome. It is not present in the inoculum and only appears in infected cells (*Chen et al., 1986*). As shown in *Figure 4A* (upper-middle panel), both HDV genomic and antigenomic RNA copies were greatly reduced in cells transfected with tsNTCP-specific siRNA but not the control siRNA, indicating that tsNTCP is required for de novo HDV infection. By contrast, lenti-VSV-G virus infection, for which viral entry is mediated by glycoprotein protein G of VSV, was not affected in the tsNTCP- and siRNA-transfected cells (*Figure 4A*, upper-right). These data demonstrate that HDV viral entry requires NTCP. As HDV is enveloped by HBV envelope proteins and can only infect target cells in a single round in the absence of HBV, these data support that tsNTCP functions at entry level for viral infection mediated by HBV envelope proteins.

We then tested HBV infection on tsNTCP knockdown PTHs. Infection with HBV can be assessed by measuring secreted viral antigens HBV S antigen (HBsAg) and HBV e antigen (HBeAg). HBV inocula may contain residual HBsAg that can release and interfere with the detection of newly synthesized HBsAg during the first few days of infection. To differentiate de novo HBsAg synthesis from the contaminating inoculum, we assayed HBsAg secretion over time from days 6 to 12 after infection with the culture medium changed every 2–3 days. In addition, the kinetics of production of HBeAg with minimal or no residuals in the inoculum was also examined in the same time course experiment. As shown in *Figure 4A* (lower-left), both HBsAg and HBeAg levels were markedly reduced by transfection of tsNTCP-specific but not a control siRNA at all three time points tested, demonstrating that tsNTCP expression is required for bona fide HBV infection. To confirm that tsNTCP functions at the viral entry level for HBV as it does for HDV, we tested AAV8-HBV virus infection on tsNTCP knockdown PTHs. AAV8-HBV is a recombinant adenovirus-associated virus containing a 1.05× overlength HBV genome, for which viral entry is mediated by AAV8 capsid instead of HBV envelope proteins. AAV8-HBV infection of PTHs can nevertheless transduce the HBV genome into cells and lead to subsequent HBV viral antigen expression. NTCP knockdown did not affect AAV8-HBV infection in PTHs, as shown by the kinetics of HBeAg (*Figure 4A*, lower-right). This result shows that NTCP has no effect on post-entry steps of HBV infection.

We next examined the effect of silencing human NTCP on HBV and HDV infections in human hepatocytes. Human hepatoma cell line HepaRG is the only cell line known to date to be susceptible to HBV and HDV infections upon differentiation into a mixture of hepatocyte-like and biliary-like cells (*Gripon et al., 2002*). HepaRG differentiation requires a lengthy cell culture procedure, including maintaining undifferentiated cells for 2 weeks before induction, followed by induction with corticoids and DMSO for another 2–4 weeks (*Gripon et al., 2002*). The NTCP mRNA level was low in HepaRG cells before induction when examined on days 5 and 10 after initial plating, but increased dramatically when the cells differentiated after induction (*Figure 4B*, upper-left). To examine if the acquired hNTCP expression on differentiated HepaRG cells is required for HDV and HBV infections, the cells were transfected with siRNAs targeting hNTCP. About 70% HDV infection was reduced by hNTCP knockdown as indicated by decreased levels of HDV viral RNAs (*Figure 4B*, upper-right). Similarly, HBV infection was also inhibited as indicated by significantly reduced HBeAg at multiple time points (*Figure 4B*, lower-left), as well as viral RNAs including the 3.5 kb RNA for HBV pre-C and pregenome RNA (pgRNA) and HBV total RNA (*Figure 4B*, lower-right) quantified at the end of the experiment. We further validated the critical role of hNTCP on HBV infection in PHHs, the natural host of the virus. Consistently, knockdown of hNTCP significantly reduced HBV infection, which was correlated with the NTCP mRNA knockdown efficiency. Both viral antigens and viral RNAs were decreased in cells transfected with hNTCP-specific siRNAs but not with the control siRNA (*Figure 4C*). Taken together, these data demonstrate NTCP as a common key cellular receptor component necessary for HBV and HDV infections of hepatocytes.

## NTCP expression renders nonsusceptible hepatocarcinoma cells susceptible to HDV and HBV infections

We then investigated the ability of NTCP to render nonsusceptible cells susceptible to viral infection. NTCP mRNA expression is low in human hepatocarcinoma cell lines that are not susceptible to HBV or

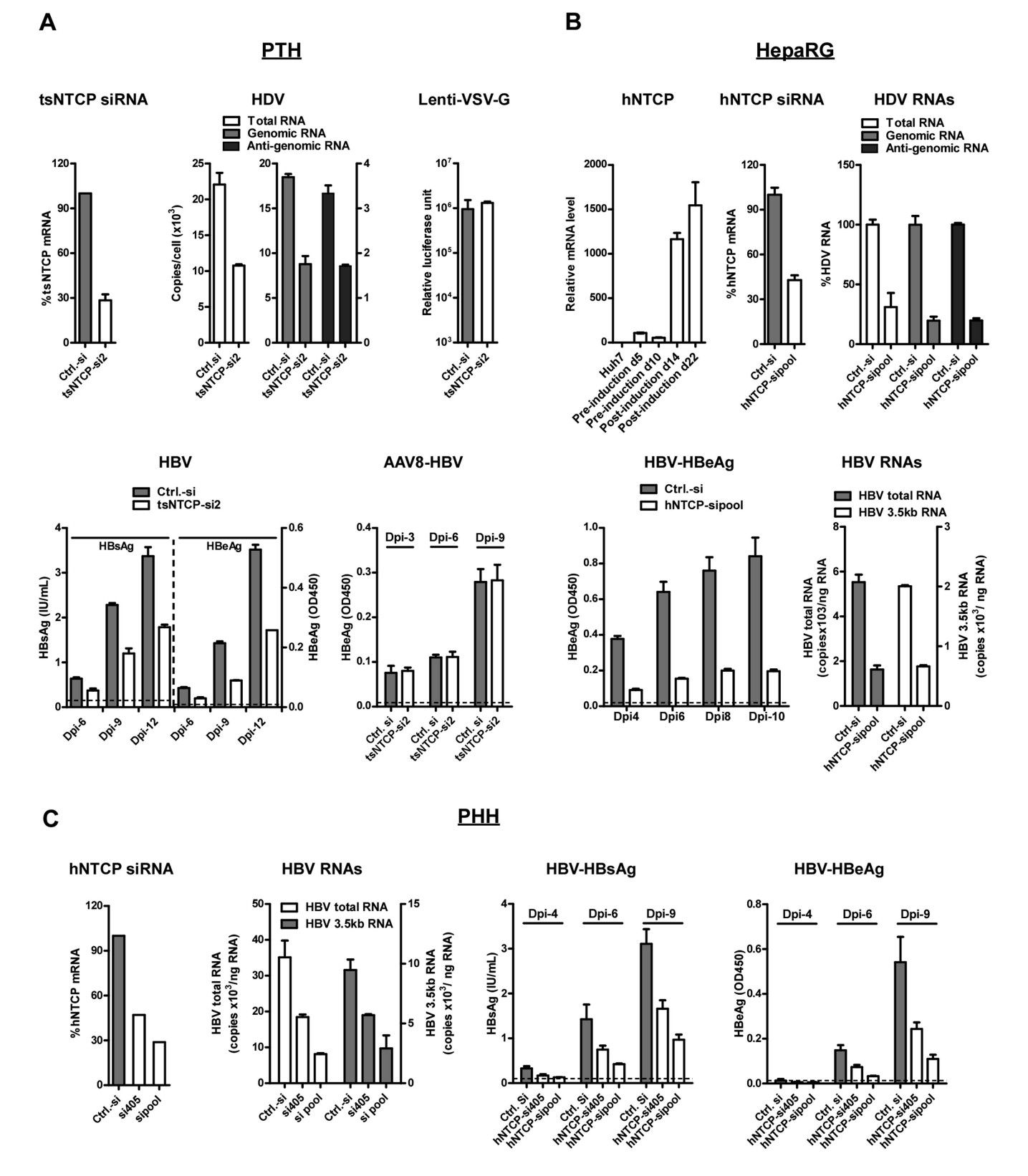

**Figure 4**. HDV and HBV infections of hepatocytes require NTCP. (**A**) Infections of HDV and HBV in PTHs were inhibited by tsNTCP knockdown. Freshly isolated PTHs were transfected with siRNAs against tsNTCP or a control siRNA. 3 days later, 1 × 10⁵ PTHs were inoculated with HDV, HBV, or control

*Figure 4. Continued on next page*

*Figure 4. Continued*

viruses AAV8-HBV and Lenti-VSV-G. For HDV and HBV, PTHs were infected at 500 and 100 genome equivalent copies per cell, respectively. The level of HDV viral RNAs in infected cells was quantified by qRT-PCR on 6 dpi. Strand-specific primers were used to differentiate the HDV genomic and anti-genomic RNAs (see 'Materials and methods'). For VSV-G control virus infection, recombinant lentivirus pseudotyped by VSV-G carrying a luciferase reporter was inoculated to PTHs 3 days after siRNA transfection. The luciferase activity was assessed on 6 dpi. For HBV infection, the kinetics of secreted viral antigens HBsAg and HBeAg were measured by ELISA. The medium was changed every 3 days. For AAV8-HBV infection, PTHs were infected with a recombinant AAV8 carrying 1.05× overlength HBV genome. Secreted HBeAg was assessed on indicated days post-infection. The effect of tsNTCP silencing in all viral infections was independently evaluated with a total of four siRNAs against tsNTCP (see 'Materials and methods'). The data shown are the result of a representative siRNA out of the four tested. (**B**) Differentiated HepaRG cells express high level of NTCP mRNA and knockdown NTCP in these cells inhibited HDV and HBV infections. HDV and HBV infection of siRNA-transfected HepaRG cells was conducted similarly as in panel A. HDV RNA levels in the infected cells were measured on 9 dpi. For HBV infection, secreted HBeAg was collected every 2 days as indicated and analyzed by ELISA. The copy numbers of HBV total RNA and 3.5 kb RNA in the infected cells were measured at the end of the experiment, 10 dpi. (**C**) Knockdown hNTCP in PHHs hampered HBV infection. Frozen PHHs were thawed and plated 1 day before transfecting with siRNAs against hNTCP or a control siRNA. Similar to panels A and B, 3 days after transfection, PHHs were inoculated with 100 genome equivalent copies of HBV per cell, and the levels of secreted HBeAg were determined at indicated dpi. HBV RNAs were quantified at the end of the experiment, 9 dpi. The knockdown efficiency of siRNA targeting tsNTCP or hNTCP shown in panels A–C was determined by real time RT-PCR on day 4 after transfection. NTCP: sodium taurocholate cotrans-porting polypeptide; HBV: hepatitis B virus; HDV: hepatitis D virus; PTH: primary *Tupaia* hepatocytes; tsNTCP: *Tupaia* NTCP; siRNA: small interfering RNA; dpi: days post-infection; hNTCP: human NTCP.

HDV infection. The levels of NTCP mRNA in Huh-7 and HepG2 cells were about 10,000 times lower than that in primary human and *Tupaia* hepatocytes (***Figure 5A***). We first examined if NTCP expression renders Huh-7 susceptible to HDV infection. Human NTCP-transfected Huh-7 cells supported HDV infection with an efficiency comparable to that of PTHs; nearly 10% of cells were infected as shown by staining of the HDV delta antigen that mainly locates in cell nuclei, whereas Huh-7 cells transfected with a vector plasmid allowed no HDV infection (***Figure 5B***). Moreover, the infection could be blocked by known HBV entry inhibitors, such as pre-S1 lipopeptide and hepatitis B immune globulin (HBIG), dem-onstrating a genuine infection of HDV mediated by HBV envelope proteins on these cells (***Figure 5C***). HDV RNAs, including antigenomic RNA that is only produced during HDV replication, rapidly increased over time in the infected cells (***Figure 5D***). Moreover, the infection efficiency correlated with both the inoculation dose of HDV (***Figure 5E***) and the expression level of hNTCP (***Figure 5F***). HDV also infected HepG2 cells transiently transfected with hNTCP (***Figure 5—figure supplement 1***) as well as a cell line established by G418 selection of HepG2 cells after hNTCP transfection, which expresses hNTCP stably and could be readily stained by the FITC-pre-S1 peptide (***Figure 5—figure supplements 2 and 3***).

Although HDV is an accepted surrogate for HBV entry, we further examined if exogenous expres-sion of NTCP directly renders nonsusceptible cells susceptible to HBV infection. HepG2 cells tran-siently transfected with either tsNTCP- or hNTCP-supported HBV infection as evidenced by continuous secretion of HBeAg over time and accumulation of HBV replicative intermediate viral RNAs in the infected cells. The entry inhibitor Myr-59 blocked the infection (***Figure 6—figure supplement 1***). More efficient HBV infection was achieved on stable HepG2-hNTCP cells with about 5–10% of the cells being infected as revealed by intracellular staining of HBsAg, whereas there was no HBV infection in the parental HepG2 cells (***Figure 6A***). HBV infection in the HepG2-hNTCP stable cells was further evidenced by the continuously increased production of HBeAg during the testing period. HBV entry inhibitors, in particular the Myr-59 peptide and 17B9, efficiently inhibited the infection (***Figure 6B***). The infection efficiency as evidenced by HBV total and 3.5 kb RNA levels correlated with the inocula-tion dose. Moreover, the formation of HBV covalently closed circular DNA (cccDNA), which is a replica-tive intermediate and transcriptional template for production of viral RNAs, was confirmed by Southern blot analysis (***Figure 6D***). To further demonstrate that the replicative intermediates of HBV were syn-thesized de novo in HepG2-hNTCP cells after infection, we performed additional time course experi-ments. HBV viral replicative intermediates, including cccDNA, the 3.5 kb HBV RNA, as well as the total HBV RNA in the HBV-infected HepG2-hNTCP cells were quantified at different time points. The cccDNA became detectable at 24 hr post-infection. It markedly increased at day 3 post-infection and maintained a relatively stable level for the rest of the time points examined, whereas the formation of HBV cccDNA was completely abolished if entry inhibitor Myr-59 was included with the initial virus inoculation (***Figure 6E***). Consistently, HBV RNA levels of the 3.5 kb transcript and the total HBV

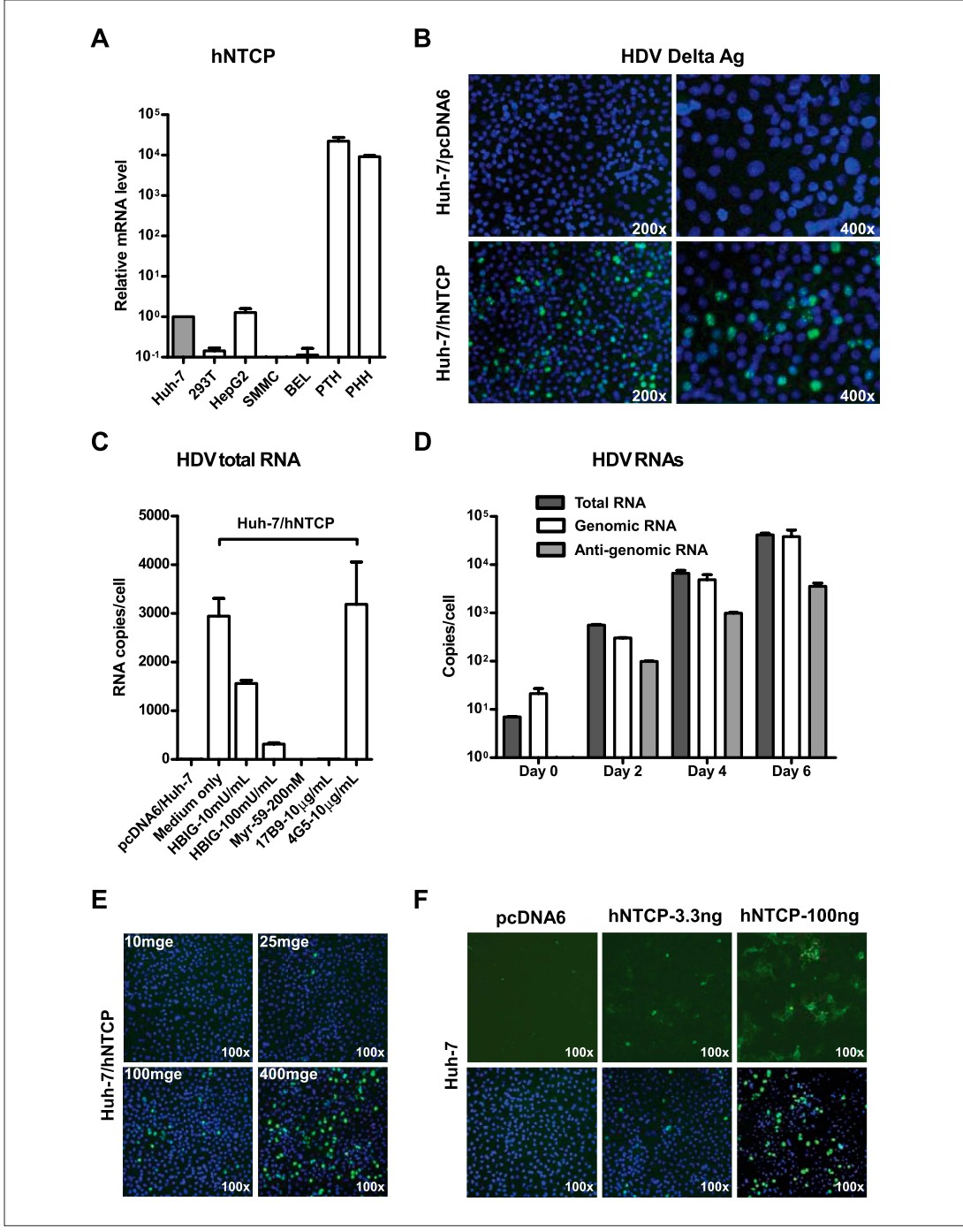

**Figure 5**. NTCP expression confers Huh-7 susceptibility to HDV infection. (**A**) NTCP mRNA expression level in the indicated cell lines and primary hepatocytes. The Huh-7 was used to normalize the relative expression levels in other cells. (**B**) $1 \times 10^5$ Huh-7 cells were transfected with 100 ng hNTCP/pcDNA6 or a vector control in 24-well plate and maintained in PMM, 24 hr after transfection, transfected cells were infected with HDV at 500 genome equivalent copies per cell. On 8 dpi, HDV delta antigen, which typically locates in nuclei, was stained with 4G5 antibody in green, nuclei were stained with DAPI in blue. (**C**) Huh-7 cells transfected with hNTCP were infected with HDV similarly as in panel B in the presence or absence of HBV entry inhibitors: HBIG (hepatitis B immune globulin), Myr-59, and anti-HBsAg mAb, 17B9. 4G5 was used as an antibody control. HDV RNA copies of infected cells were quantified by real-time RT-PCR on 6 dpi. (**D**) Huh-7 cells transfected with hNTCP were infected with HDV similarly as in panel B. The HDV viral RNAs in infected cells at indicated time points were quantified by real-time RT-PCR. (**E**) HDV infection with increasing multiplicities of genome equivalents (mge). With 100 ng hNTCP/pcDNA6, $1 \times 10^5$
*Figure 5. Continued on next page*

*Figure 5. Continued*

Huh-7 cells were transfected, as in panel B. Transfected cells were infected with increasing mge of HDV as indicated. HDV delta antigen was detected as in panel B on 8 dpi. (**F**) HDV infection of cells with increasing levels of hNTCP. About $1 \times 10^5$ Huh-7 cells were transfected with a vector pcDNA6 or hNTCP/pcDNA6 at indicated amounts and cells were inoculated with 500 mge of HDV. HDV delta antigen was detected on 8 dpi as in panel B. NTCP: sodium taurocholate cotransporting polypeptide; PMM: primary hepatocytes maintenance medium; HBV: hepatitis B virus; HDV: hepatitis D virus; mAb: monoclonal antibody; HBsAg: HBV S antigen.

The following figure supplements are available for figure 5.

**Figure supplement 1**. Infection of HDV on HepG2 cells expressing hNTCP.

**Figure supplement 2**. Infection of HDV on HepG2 cells expressing hNTCP.

**Figure supplement 3**. Infection of HDV on HepG2 cells expressing hNTCP.

transcripts in the infected HepG2-hNTCP cells gradually increased during first several days of infection and reached a steady level after day 5 (*Figure 6F*). Together these data show that NTCP contributes substantially to HBV infection.

We next compared the efficiency of HBV infection in HepG2-hNTCP cells with that in PHHs. As shown by intracellular staining of HBV core antigen (HBcAg) on day 8 post-infection, about 10% HepG2-NTCP cells were infected at multiplicities of genome equivalents (mge) of 100, which is comparable to the efficiency of HBV infection of PHHs (*Figure 6—figure supplement 2*). In contrast to PHHs, HepG2-NTCP cells propagate in cultures, thus the actual infection efficiency of HepG2-NTCP cells may be more likely than not underestimated by the observed end-point HBcAg staining. We also compared the levels of secreted HBeAg and intracellular viral RNAs in these two types of cells infected with three inoculation doses. The level of secreted HBeAg from HepG2-NTCP appeared to be higher than that in PHHs from two donors, whereas the levels of viral RNAs per nanogram of total cell RNA in both infected cell types are comparable (*Figure 6—figure supplement 3*). This may be partially explained by their different abilities in propagation and supporting viral replication and protein expression that would require more detailed studies.

Efficient HBV infection of PHHs or HepARG cells in vitro normally requires high dose of virus inoculums, and only limited progeny viruses are produced after infection (*Gripon et al., 1988*; *Gripon et al., 2002*; *Boehm et al., 2005*). To assess viral particles released from HBV-infected HepG2-NTCP cells, we first quantified viral DNA in the medium collected at different time points after the infection. As indicated by drastic decline of viral DNA level on day 4 post-infection, the majority of residual viruses from inocula were removed by changing the medium and washing during the first few days of infection. The levels of viral DNA in the media resulted from the ongoing infection during days 4–13 post-infection were low (equivalent to ~1% of input viral DNA copies) despite significant amount of HBeAg secretion during this period. Similarly, only low levels of viral DNA were detected in the medium from HBV-infected PHHs (*Figure 6—figure supplement 4*, right). It is reasonable to speculate that some host factors that are lacking in cell cultures might be needed for efficient viral particles formation or releasing; or some cellular factors in cultures may hinder these processes during infection. The culture medium collected from infected HepG2-NTCP cells was subsequently tested for infection of PHHs. In line with the low HBV viral DNA level in the medium inoculum, very low number of intracellular HBV total RNA copies were detected in PHHs on day 13 post-infection (*Figure 6—figure supplement 5*), indicating that only very limited HBV infection might have occurred, which may be attributed to the low multiplicity of infection.

## Residues 157 to 165 of hNTCP are critical for pre-S1 binding and viral infections

We finally investigated the molecular determinants of NTCP for HBV and HDV infections. Crab-eating monkey (*Macaca fascicularis*) NTCP (mkNTCP) shares high protein sequence identity with hNTCP (96.3%) (*Figure 7—figure supplement 1*). However, mkNTCP neither supports HDV infection nor pre-S1 peptide (Myr-59) binding (*Figure 7A*), consistent with the known narrow species specificity of the

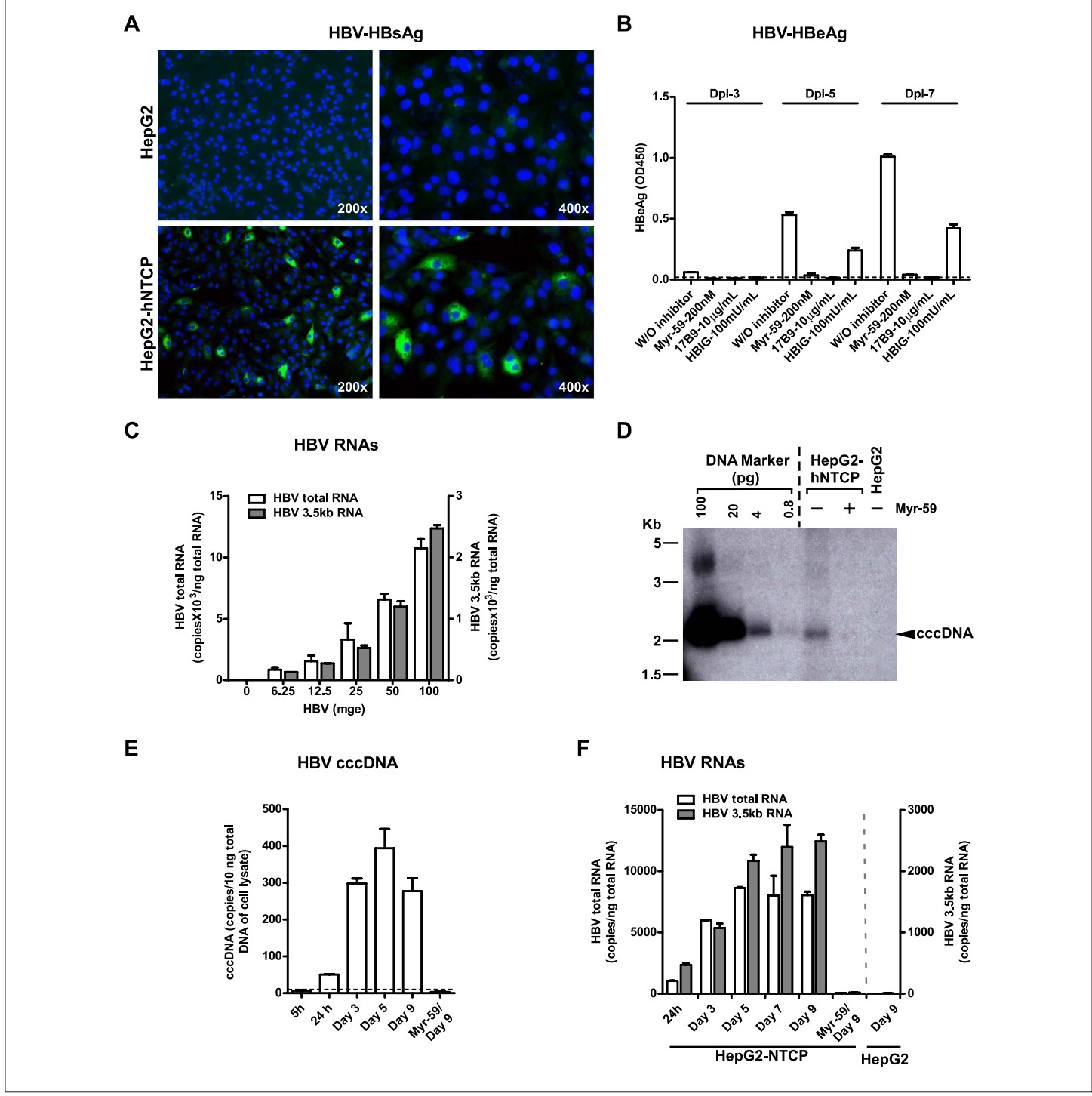

**Figure 6**. NTCP expression confers susceptibility to HBV infection. (**A**) Intracellular HBsAg expression in HBV-infected cells. HepG2-hNTCP stable cells or parental HepG2 cells were inoculated with HBV at 100 mge. On 9 dpi, intracellular HBsAg of infected cells on coverslips was stained with antibody 17B9 in green, and nuclei were stained with DAPI in blue. (**B**) Secreted HBeAg levels in the supernatants of HBV-infected cells. The cells were infected with HBV at 100 mge in the presence or absence of entry inhibitors as indicated. The medium was changed every 2 days. Secreted HBeAg was measured at 3, 5, 7 dpi; each time point represents the level of newly synthesized HBeAg within every 2 days. (**C**) HBV infection efficiency is correlated with the viral inoculum dose. With increasing dose of HBV, $2 \times 10^5$ HepG2-hNTCP cells were infected as indicated. HBV RNAs in infected cells was examined on 10 dpi with real-time RT-PCR for the total HBV RNAs and the 3.5 kb transcripts. (**D**) Southern blot analysis of cccDNA. HepaG2-hNTCP cells or HepG2 parental cells were infected with 100 mge of HBV in the presence or absence of Myr-59. HBV cccDNA was extracted from ~ 2–3 × 10^6 infected cells on 7 dpi. Half of the extracted DNA of each sample was subjected to a 1.3% agarose gel and analyzed by Southern blotting. A plasmid DNA marker for cccDNA (see 'Materials

*Figure 6. Continued on next page*

*Figure 6. Continued*

and methods') at different concentrations from 0.8 to 100 pg was included in the same gel. (**E**)–(**F**) Kinetic analysis of HBV cccDNA and RNAs in HBV-infected HepG2-hNTCP cells. HBV cccDNA (in panel E) was quantified at indicated time points post infection (see 'Materials and methods'). A dotted line indicates the background amplification. HBV RNA copies (in panel F) in the infected cells were measured at indicated time points, data of similarly infected parental HepG2 cells on 9 dpi were also shown. NTCP: sodium taurocholate cotransporting polypeptide; HBV: hepatitis B virus; HBsAg: HBV S antigen; HBeAg: HBV e antigen; mge: multiplicities of genome equivalents; dpi: days post-infection; cccDNA: covalently closed circular DNA; hNTCP: human NTCP.

The following figure supplements are available for figure 6.

**Figure supplement 1**. NTCP expression confers susceptibility to HBV infection.

**Figure supplement 2**. Comparative HBV infection of PHHs and HepG2-NTCP cells.

**Figure supplement 3**. Comparative HBV infection of PHHs and HepG2-NTCP cells.

**Figure supplement 4**. Kinetics of HBV viral DNA in the culture medium of primary infection and reinfection of the released viruses on PHHs.

**Figure supplement 5**. Kinetics of HBV viral DNA in the culture medium of primary infection and reinfection of the released viruses on PHHs.

virus. We then made a series of human NTCP variants to cover all the different amino acids between hNTCP and mkNTCP. In each variant, two or a few residues were mutated to their mkNTCP counterparts (*Figure 7—figure supplement 1*). Whereas most mutations did not significantly interfere with Myr-59 binding or HDV infection, alteration of five residues of hNTCP between aa 157–165 and its monkey counterpart (from <u>KGIVI</u>SLV<u>L</u> to <u>GRIIL</u>SLV<u>P</u>, distinct residues are underlined) completely abolished Myr-59 binding and the ability to support HDV infection. Remarkably, replacing the motif of aa 167–156 in mkNTCP with the corresponding human residues converted mkNTCP to an efficient receptor for HDV infection (*Figure 7A*). All the NTCP variants tested were examined for NTCP expression, and comparable levels of cell surface expression were confirmed (*Figure 7B*). Similar to HDV, HBV infection was also abolished on HepG2 cells expressing hNTCP carrying monkey-like mutations, <u>GRIIL</u>SLV<u>P</u>, while mkNTCP-bearing human residues <u>KGIVI</u>SLV<u>L</u> within the motif of aa 157–165 largely restored HBV infection (*Figure 7C*). These data show that residues between 157 and 165 of NTCP are crucial for binding to the receptor-binding region of the pre-S1 domain of the L protein of HBV, and critically contribute to NTCP-mediated HBV and HDV infections.

## Discussion

In this study, by employing a unique approach of tandem affinity purification combined with MS analysis against a *Tupaia* hepatocyte proteome database established by deep sequencing, we revealed that the liver bile acid transporter, NTCP, specifically interacts with a key region in the pre-S1 domain of the HBV envelope L protein. By performing a series of virological analyses, we showed that silencing NTCP expression markedly inhibited viral infection of HBV and HDV in *Tupaia* as well as human hepatocytes. Exogenous expression of NTCP rendered nonsusceptible human hepatoma cells susceptible to the viral infections. The authentic viral infections in cells complemented with NTCP were shown by the kinetic analyses of several markers of viral infections, in particular the quantification of newly synthesized viral replicative intermediates. Moreover, the NTCP-rendered infections were blocked by known entry inhibitors. NTCP residues 157 to 165 were identified to be critical for pre-S1 binding and viral infections. These data clearly demonstrate that NTCP is a functional receptor for both HBV and HDV.

Identification of cellular receptor(s) of HBV and HDV has been challenging. In our study, we utilized a short peptide ligand, WT$_b$, which was originated from the known receptor-binding domain of the L protein (*Barrera et al., 2005*; *Glebe et al., 2005*; *Gripon et al., 2005*; *Engelke et al., 2006*; *Schulze et al., 2010*), but with specially designed properties suitable for photo-cross-linking and tandem purification.

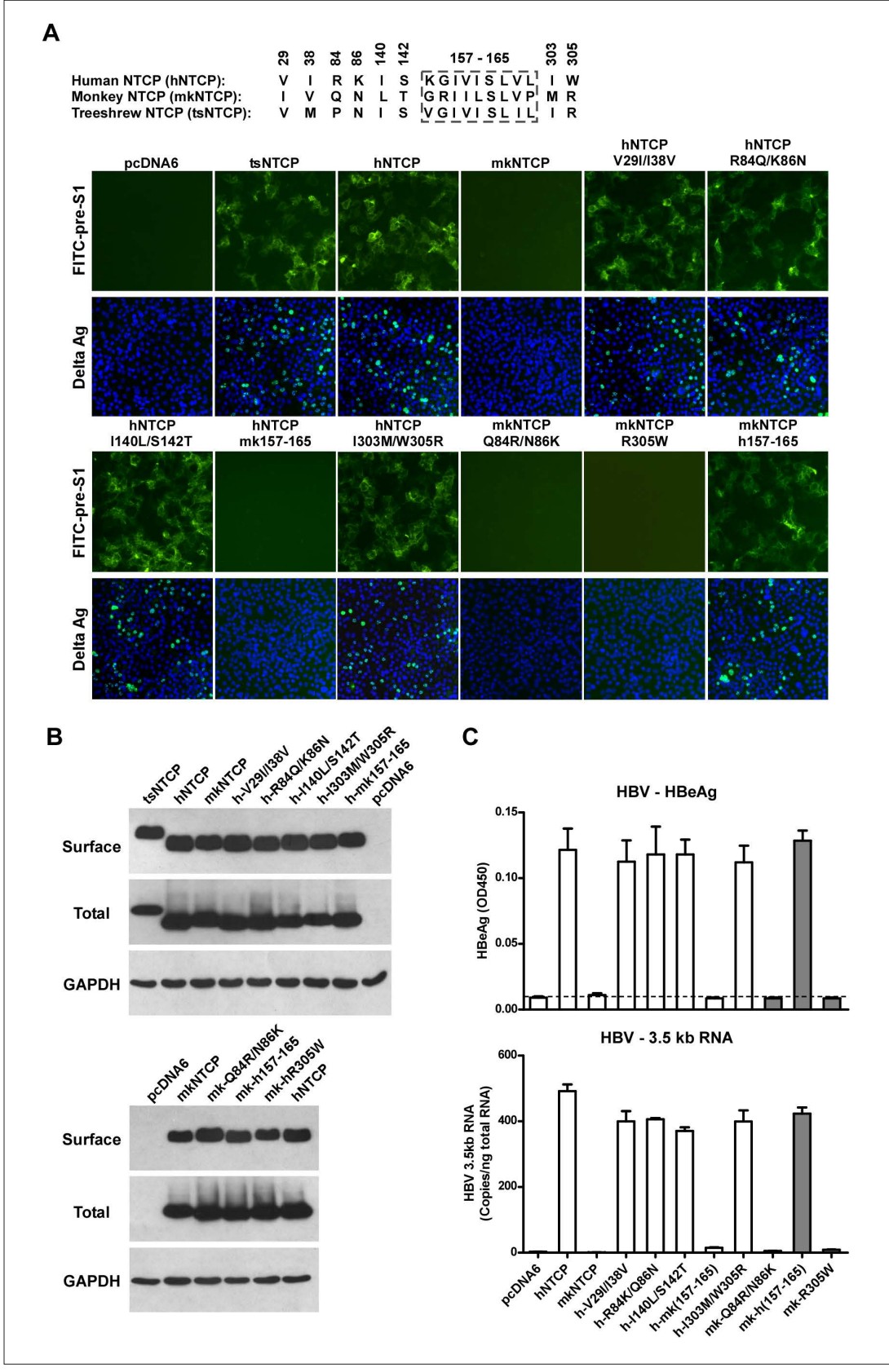

**Figure 7**. Identification of a critical region (aa 157–165) of NTCP for pre-S1 binding and viral infections. (**A**) Pre-S1 binding and HDV infection on cells expressing wild-type or mutant NTCPs. Corresponding amino acids (one-letter

*Figure 7. Continued on next page*

*Figure 7. Continued*

form) at the mutated positions of NTCP are shown for hNTCP, crab-eating monkey NTCP (mkNTCP), and tsNTCP. Huh-7 cells were transfected with plasmids encoding tsNTCP, hNTCP, mkNTCP, or NTCP mutants as indicated. The mutant NTCPs include hNTCP-bearing mutations of mkNTCP residues and mkNTCP-bearing mutations of human residues at indicated positions. The transfected cells were maintained in PMM for 24 hr and then either stained with 200 nM FITC-pre-S1 or infected with 500 mge HDV. under the same experimental conditions as described in *Figure 5B*. HDV delta antigen in infected cells was detected with mAb 4G5 on 8 dpi. The delta antigen staining images of pcDNA6 and hNTCP shared the same source images for Huh-7/pcDNA6 and Huh-7/ hNTCP in *Figure 5B*, respectively. Replacing aa 157–165 of mkNTCP with human counterpart rendered mkNTCP an efficient receptor for pre-S1 binding and HDV infection. (**B**) All NTCP variants expressed comparable levels of NTCP. Huh-7 cells transfected as in panel A were biotinylated 24 hr after the transfection, then lysed and analyzed for cell surface NTCP expression (top), total NTCP expression (middle), and GAPDH (bottom), respectively. For cell surface expression, cell lysates were pulled down with streptavidin T1 Dynabeads and subsequently examined by western blot with mAb 1D4 recognizing a C9 tag at the C-terminus of each NTCP variant. For total NTCP expression, cell lysates were directly subjected to SDS-PAGE, followed by Western blot analysis with 1D4. (**C**) Effects of NTCP mutations on HBV infection. HepG2 cells were transfected with plasmids encoding hNTCP, mkNTCP, or hNTCP variants bearing the indicated monkey residues, or mkNTCP variants with the indicated human residues. Transfected cells were maintained in PMM for 24 hr, and subsequently infected with HBV at 100 mge. HBeAg and HBV 3.5 kb RNA were assayed on 6 dpi. Similar to panel B, comparable NTCP surface expression levels in the transfected HepG2 cells were confirmed for all the NTCP variants tested (***Figure 7—figure supplement 2***). NTCP: sodium taurocholate cotransporting polypeptide; HDV: hepatitis D virus; hNTCP: human NTCP; tsNTCP: *Tupaia* NTCP; PMM: primary hepatocytes maintenance medium; mge: multiplicities of genome equivalents; mAb: monoclonal antibody; HBV: hepatitis B virus; HBeAg: HBV e antigen; dpi: days post-infection.

The following figure supplements are available for figure 7.

**Figure supplement 1**. Protein sequence alignment of human, monkey, and *Tupaia* NTCP. Residues different between human and monkey are highlighted in red. *Tupaia* residues different from human's are in blue. Residues 157–165 are boxed.

**Figure supplement 2**. Total and surface NTCP expression levels in the transfected HepG2 cells. Transiently transfected HepG2 cells from the same batch of transfection as in ***Figure 7C*** were analyzed for total or cell surface NTCP expression at 24 hr post-transfection as described in ***Figure 7B***.

---

Two photo-leucines were incorporated into the critical receptor-binding region of $WT_b$ without interfering with its receptor-binding activity, which allowed highly specific zero distance cross-linking of its direct binding partner(s) but not other neighboring molecules. A biotin moiety of $WT_b$ facilitated purification of the complex of $WT_b$ and its binding partner(s) by streptavidin beads. An mAb, 2D3, was developed to recognize $WT_b$ on an epitope outside the receptor-binding site, serving as a highly specific tool for detection as well as additional affinity purification of the complex. Thus, the binding partner(s) was first cross-linked by $WT_b$, and then purified by using streptavidin and 2D3 beads in tandem. The covalent interaction between the $WT_b$ ligand and its partner(s) enabled a purification process under high-stringency conditions, and efficient isolation was achieved irrespective of the nature of the binding partner(s) even if it is a membrane protein(s) with multiple transmembrane domains, like NTCP identified here.

NTCP (*Slc10a1*) is the founding member of the SLC10 family of solute carrier proteins. It is a hepatic $Na^+$ bile acid symporter and is responsible for cotransportation of sodium and bile acids across cellular membranes to maintain the enterohepatic circulation of bile acids (***Hagenbuch and Meier, 1994***; ***Stieger, 2011***). NTCP is a multiple transmembrane glycoprotein presumed to span the cellular membrane up to 10 times with small extracellular loops (***Mareninova et al., 2005***; ***Hu et al., 2011***). It is mainly expressed in the liver (***Stieger, 2011***), consistent with the liver tropism of HBV and HDV. NTCP localizes to the sinusoidal (basolateral) plasma membrane of hepatocytes (***Stieger et al., 1994***), a location that fits well with its receptor role for blood-borne HBV and HDV. Whereas HBV first attaches to hepatocytes mainly through heparan sulfate (***Schulze et al., 2007***; ***Leistner et al., 2008***), our data demonstrate that the interaction between NTCP and L protein of HBV is highly specific, and NTCP is crucial for productive viral entry of hepatocytes. Consistent with previous reports on primary cultures of rat hepatocytes (***Liang et al., 1993***; ***Rippin et al., 2001***), NTCP expression rapidly decreased over time in cultured PTHs after isolation. This may at least partially explain the observations that primary

hepatocytes typically remain susceptible to HBV infections in vitro for only a few days after isolation from liver tissues (*Gripon et al., 1988*; *Seeger et al., 2007*).

NTCP is functionally conserved in mammalians, but protein sequences of NTCP vary among species, which is likely to contribute to the narrow species tropism of viral infection. Strikingly, despite the high level of protein sequence homology between human and monkey NTCP, the later did not support HBV and HDV infection. Replacing a small motif of aa 157–165 of mkNTCP with the corresponding hNTCP residues converted mkNTCP to a receptor for pre-S1 binding as well as HDV and HBV infection. Further studies are warranted to determine if and how NTCP contributes to the species specificity of HDV and HBV infection in other species. It also remains to be determined if other molecule(s) additional to NTCP contributes to the cellular entry of HBV and/or HDV as a coreceptor(s) or receptor component(s), and if other host factors such as the microenvironment or architecture of hepatocytes in liver, or soluble blood components like those that have been shown to involve in infections of other viruses (*Shayakhmetov et al., 2005*; *Morizono et al., 2011*), contribute to HBV and/or HDV infection.

Expression and subcellular distribution of NTCP are precisely regulated under physiological conditions. NTCP accounts for most, if not all, hepatic Na$^+$-dependent bile acid transport (*Stieger, 2011*). NTCP expression is low and inversely correlated with the degree of dedifferentiation of cancer cells in human hepatocellular carcinoma (*Kullak-Ublick et al., 1997*; *Zollner et al., 2005*) and the severity of HBV-related liver cirrhosis (*Lee and Kim, 2007*). The newly discovered role of NTCP as an entry receptor for HBV and HDV raises interesting questions regarding its involvement in viral pathogenesis. Identification of NTCP as a functional receptor for HBV and HDV advances our understanding of their entry into host cells and may lead to new prevention and treatment strategies against these viruses and related diseases.

## Materials and methods

### Isolation and culture of primary *Tupaia* hepatocytes (PTHs)

Adult tree shrews (*Tupaia belangeri chinensis*) were housed in a *Tupaia* animal facility at the National Institute of Biological Science, Beijing. All studies were performed in accordance with institutionally approved protocols and adherent to guidelines of the National Institute of Biological Sciences Guide for the care and use of laboratory animals. PTH cells were obtained from anesthetized *Tupaia* (100–150 g) with a two-step perfusion method as previously described (*Walter et al., 1996*). Cell suspensions after perfusion were filtered through a 70-µm cell strainer and centrifuged at 50 *g* for 3 min. The cell pellet containing PTHs was resuspended in plating medium of Williams E medium supplemented with 10% FBS, 5 µg/ml transferrin, 5 ng/ml sodium selenite, 2 mM L-glutamine, 100 U/ml penicillin, and 100 µg/ml streptomycin. The cells were then plated on collagen-coated cell culture dishes or plates. 4 hr after plating, medium were changed to primary hepatocytes maintenance medium (PMM), that is, Williams E medium supplemented with 5 µg/ml transferrin, 10 ng/ml EGF, 3 µg/ml insulin, 2 mM L-glutamine, 18 µg/ml hydrocortisone, 40 ng/ml dexamethasone, 5 ng/ml sodium selenite, 2% DMSO, 100 U/ml penicillin, and 100 µg/ml streptomycin. Cells were maintained in 5% CO$_2$ humidified incubator at 37°C with regular medium change every 2–3 days.

### Primary human hepatocytes (PHHs)

PHHs were purchased from Becton Dickinson (United States) or Shanghai RILD Inc. (Shanghai, China). The cells were cultured similarly as PTHs using the same plating medium and maintaining PMM medium as described above.

### Cell lines

Human embryonic kidney cell lines 293 and 293T, human cervix carcinoma cell line Hela, and human hepatocellular carcinoma cell line HepG2 were from American Type Culture Collection (ATCC); human hepatocellular carcinoma cell lines Huh-7, SMMC-7721 (SMMC), and Bel-7404 (BEL) were from the Cell Bank of Type Culture Collection, Chinese Academy of Sciences. The cells were cultured with Dulbecco's Modification of Eagle's Medium (DMEM; Invitrogen, United States) supplemented with 10% fetal bovine serum, 100 U/ml penicillin, and 100 µg/ml streptomycin at 37°C in 5% CO$_2$ humidified incubator except otherwise indicated. HepaRG cells were purchased from Biopredic International

(Rennes, France) and were cultured following the product manual. Differentiated HepaRG cells were obtained following a two-step procedure as described by *Gripon et al. (2002)*.

## Viruses

### HDV

A plasmid containing a head to tail trimer of 1.0× HDV cDNA of a genotype I virus (Genebank accession number: AF425644.1) under the control of a CMV promoter was constructed with de novo synthesized HDV cDNA for the production of HDV RNPs. A pUC18 plasmid containing nucleotide 2431–1990 of HBV (Genotype D, Genebank accession number: U95551.1), or the same plasmid bearing mutation generated by site-directed mutagenesis, was used for expressing HBV envelope proteins under the control of endogenous HBV promoter. HDV virions were produced by transfection of the plasmids in Huh-7 as previously described by *Sureau et al. (1992)*.

### HBV

HBV genotype B virus was obtained by ultracentrifugation of plasma from an HBV chronic carrier with written consent. HBV genotype D virus was produced by transfection of Huh-7 cells with a plasmid containing 1.05 copies of HBV genome under the control of a CMV promoter similarly as previously described by *Blanchet and Sureau (2006)*. The Genebank accession numbers for the viruses are JX978431 and U95501.1, respectively.

### AAV8-HBV recombinants

Recombinant adeno-associated virus 8 (AAV8) carrying 1.05 copies of HBV genome was produced similarly as previously described (*Xiao et al., 1998*) by cotransfection of 293 cells with plasmids for AAV8 packaging, 1.05× overlength HBV genome (genotype D) and adenovirus helper.

### Lenti-VSV-G

An HIV-1 genome-based lentivirus pseudotyped by glycoprotein of vesicular stomatitis virus and carrying a firefly luciferase reporter gene was produced by cotransfection of 293T cells with plasmids for VSV-G expression, HIV genome packaging, and luciferase reporter, respectively, as described (*Sui et al., 2005*).

Virus-related experiments were conducted in a BSL-2 facility at the National Institute of Biological Sciences, Beijing.

## Peptides and antibodies

Peptides with nonnatural amino acid *L-2-amino-4,4-azi-pentanoic acid* (L-photo-leucine) were synthesized by American Peptide Company Inc. (United States). Other peptides corresponding to the N-terminal of pre-S1 domain of HBV L protein (genotype C, strain S472, GeneBank EU554535.1) were synthesized by SunLight Peptides (Beijing, China). Mouse monoclonal antibodies (mAb) 2D3, 1C10, and 4G5 were generated in the laboratory; all are of IgG1 isotype. 2D3 specifically recognizes the 19–33 amino acids of the pre-S1 domain of HBV L protein; 1C10 recognizes HBcAg; 4G5 targets HDV delta antigen. 17B9, a mouse mAb-specific to HBV S protein, was provided by Dr. Lin Jiang, China National Biotec Group. Hepatitis B immune globulin (HBIG) was from the National Institutes for Food and Drug control, Beijing, China. 2D3 magnetic beads were prepared by covalently cross-linking 2D3 to Dynabeads M-270 Epoxy following manufacturer's instructions. Secondary antibodies for immunofluorescence staining and Western blot were purchased from Invitrogen or Sigma-Aldrich (United States).

## ELISA kits and other reagents

ELISA kits for HBsAg and HBeAg measurement were purchased from Wantai Pharm Inc. (Beijing, China). SYBR Premix Ex Taq quantitative real-time PCR kit and Reverse Transcriptase (RT) kit were from Takara Inc. (Beijing, China). Streptavidin-coupled magnetic beads (Dynabeads MyOne Streptavidin T1) and magnetic beads coated in glycidyl ether (Epoxy) groups (Dynabeads M-270 Epoxy) were purchased from Invitrogen. Other reagents were purchased from New England Biolabs (United States), Life Technologies (United States), or Sigma-Aldrich.

## Assays for HBV viral antigens from supernatant of infected cells

HBV viral antigens HBsAg and HBeAg were examined using 50 μl supernatants with commercial ELISA Kits (Wantai Pharmacy, Beijing, China) following manufacturer's instructions. In most cases, HBsAg

level was normalized with WHO HBsAg reference serum (kindly provided by Dr. Zhenglun Liang from the National Institutes for Food and Drug control, Beijing, China) and presented as international units per milliliter.

## Quantification of HDV total RNA (genome equivalent) copies and HBV genome equivalent copies

### HDV
Viral RNA was isolated with Trizol reagent following manufacturer's instructions. Total RNA was reverse transcribed into cDNA with random primers (PrimeScript RT kit; Takara) and 2 μl of the cDNA was used for real-time PCR assay. Primers for quantifying HDV total RNA or genome equivalent copies are complemented with the delta antigen coding region of HDV RNA genome: forward primer HDV-1184F, 5′-TCTTCCTCGGTCAACCTCTT-3′, and backward primer HDV-1307R, 5′-ACAAGGAGAGGCAGGATCAC-3′.

### HBV
Viral DNA was isolated by standard genomic DNA isolation method. The DNA was quantified using specific primers: 5′-GAGTGTGGATTCGCACTCC-3′ (forward) and 5′-GAGGCGAGGGAGTTCTTCT-3′ (backward) by real-time PCR. The viral genome equivalent copies were calculated based on a standard curve generated with known copy numbers. Real-time PCRs were performed using SYBR Premix Ex Taq kit on an ABI Fast 7500 real-time system instrument (Applied Biosystems, United States).

## HDV binding and inhibition assays
With $5 \times 10^7$ copies of genome equivalent HDV, $1 \times 10^5$ of target cells were incubated at 16°C for 4 hr in the presence of 4% PEG8000, followed by extensive wash with cold PBS for four times. The cells were then lysed directly with Trizol reagent and followed by reverse transcription. RNA copy numbers of viral genome and internal control glyceraldehyde-3-phosphate dehydrogenase (GAPDH) mRNA were determined by real-time PCR. For HDV binding inhibition assay, peptides were pre-incubated with target cells at 16°C for 1 hr before incubating with the virus; antibodies against viral envelope protein were pre-incubated with viruses before adding to target cells.

## HDV and HBV infection and inhibition assays
Viral infections of HDV and HBV were conducted in 48-well plates at multiplicities of genome equivalents of 500 and 100, respectively. Normally, $5 \times 10^7$ copies of genome equivalent HDV or $1 \times 10^7$ copies of genome equivalent HBV were inoculated in the presence or absence of entry inhibitors with $1 \times 10^5$ cells and incubated for 16 hr except otherwise indicated. Cells were then washed with medium for three times and maintained in PMM medium with medium change every 2–3 days. For HDV infection, 4% PEG 8000 was present during the 16 hours viral inoculation period similarly as described by *Barrera et al. (2004)*. Viral infection at different time points was analyzed by measuring viral DNA/RNAs and viral antigen expression. Quantitative real time RT-PCR was used to quantify HDV total RNAs, strand-specific real time RT-PCR to determine copies of HDV genome and antigenome RNA (see below). For HBV viral infection on HepaRG cells and PHH, ~ 4% PEG 8000 was present during the inoculation period as previously described by Gripon et al. (*Gripon et al., 1993*; *Gripon et al., 2002*; *Schulze et al., 2007*). Viral infection of PTH was conducted in the absence of PEG8000. Culture medium was changed every 2–3 days. Secreted HBsAg and/or HBeAg were determined with commercial ELISA kits. Real-time PCR, with or without a prior reverse transcription step, was used for quantification of HBV-specific 3.5 kb pre-C and pregenomic RNA, total HBV sub-genomic RNA, and HBV cccDNA copies.

## Quantification of HDV genome and antigenome RNAs by strand-specific real time RT-PCR
The strand-specific qRT-PCR was performed as previously described by *Freitas et al. (2012)*. Briefly, the genomic and antigenomic RNAs were reverse transcribed separately with strand-specific primers into cDNAs: primer HDV398R (5′-CGCTTCGGTCTCCTCTAACT-3′) for genomic RNA; primer HDV288F (5′-GCAGACAAATCACCTCCAGA-3′) for antigenomic RNA. The reverse transcribed cDNAs of genmomic or antigenmoic HDV RNAs were used as templates for real-time PCR using the HDV398R and HDV288F primer pair. TaqMan probe was 5′ FAM-AGAGCTCTGACGCGCGAGGAGTAAGC-TAMRA 3′. Real-time PCR assays were conducted with an ABI Fast 7500 real-time PCR instrument.

## Quantification of HBV-specific RNAs

Total RNA from HBV-infected cells was isolated with Trizol reagent (Invitrogen). About 400 ng total RNA was reverse transcribed into cDNA with PrimeScript RT kit (Takara) in a 10 µl reaction. cDNA derived from 20 ng total RNA was used as template for real-time PCR amplification. In a separate real-time PCR reaction, 20 ng of total RNA was directly used as template to assess the possible HBV viral DNA contamination in the RNA preparation. Primers (HBV2270F: 5'-GAGTGTGGATTCGCACTCC-3') and (HBV2392R: 5'-GAGGCGAGGGAGTTCTTCT-3') were used for HBV 3.5 kb transcripts; (HBV1805F: 5'-TCACCAGCACCATGCAAC-3') and (HBV1896R: 5'-AAGCCACCCAAGGCACAG-3') were for total HBV-specific transcripts. Amplification of 123-bp fragment for 3.5 kb transcripts and 92-bp product for total HBV-specific transcripts were both conducted by denaturation at 95°C for 30 s, followed by 40 cycles of 95°C denaturation for 3 s, and 60°C annealing/elongation for 30 s. Real-time PCR was performed using SYBR Premix Ex Taq kit on an ABI Fast 7500 real-time system instrument. Real-time PCR using either set of the primers generated highly specific amplification product. HBV RNA copy numbers were deduced from a standard curve generated from known nucleic acid quantities. Then the HBV RNA copy number per nanogram RNA in the infected cell cultures was calculated by subtracting the background amplification noise derived from the viral DNA contamination in the RNA preparation from that cDNA amplification. The signal-to-noise ratio for HBV total transcripts is usually ≥50, and for 3.5 kb transcripts ≥20. The real-time PCR detection limits for total HBV-specific transcripts and 3.5 kb transcripts are ~0.5 and ~3.5 copies per nanogram cellular total RNA, respectively.

## Southern blot analysis of HBV covalently closed circular DNA (cccDNA)

HBV cccDNA Southern blot was conducted following a similar procedure as described by *Summers et al. (1990)* with modifications. Briefly, to selectively extract HBV cccDNA, infected hepG2-NTCP cells in 6-cm dishes were lysed with 1 ml lysis buffer at 37°C for 60 min, followed by addition of 0.25 ml of 2.5 M KCl and incubation at 4°C overnight. The lysis buffer was not supplemented with proteinase K, containing 50 mM Tris–HCl, pH 7.4, 10 mM EDTA, 150 mM NaCl, 1% SDS. The lysate was then clarified by centrifugation at 12,000 g for 30 min at 4°C and extracted with phenol and phenol:chloroform. DNA was precipitated with equal volume of isopropanol in the presence of 20 µg glycogen (Roche) and finally dissolved in TE buffer. The prepared DNA sample was then treated with plasmid-safe adenosine triphosphate (ATP)-dependent deoxyribonuclease DNase (Epicentre Technologies) following manufacturer's instructions. For Southern blotting, the plasmid-safe DNase-treated DNA was separated on a 1.3% agarose gel and then transferred to a nylon membrane (Hybond-N+; Amersham) using a standard neutral transfer procedure. A 3280-bp plasmid constructed by inserting a 588-bp HBV DNA fragment (from 1805 to 2392, genotype D, Southern blot probe) into a 2692 bp pMD18T vector (Takara) was also run on the same agarose gel to serve as the molecular marker for cccDNA in Southern blot analysis. The plasmid is of similar size of HBV genome and was mainly in supercoiled form; therefore it runs at similar size as HBV cccDNA in agarose gel. The nylon membrane was hybridized with a [α-$^{32}$P] dCTP-labeled HBV probe (genotype D HBV DNA fragment from 1805 to 2392) prepared by random primer DNA labeling kit (Ver.2.0; Takara). Hybridization was carried out in 7 ml of Perfect Hyb Plus Hybridization Buffer (Sigma) with 1 hr pre-hybridization, followed by overnight hybridization at 67°C. The membrane was then washed once with 2× SSC/0.1% SDS, 1× SSC/0.1% SDS, and 0.5× SSC/0.1% SDS at 67°C for 20 min, respectively. Finally, the membrane was subjected to autoradiographic exposure.

## Quantification of HBV cccDNA by qPCR

HBV cccDNA (double-stranded DNA without nick and gap) was quantified by real-time PCR using a protocol as previously described (*Bowden et al., 2004*; *Werle-Lapostolle et al., 2004*) with modifications. In particular, specific primers for cccDNA detection reported by *Glebe et al. (2003)* (ccc-1582F: 5'-TGCACTTCGCTTCACCT-3'; ccc-2316R: 5'-AGGGGCATTTGGTGGTC-3') were validated and used for quantifying copy numbers of cccDNA using real-time PCR. Viral DNAs other than cccDNA, including single-stranded and relaxed circular DNAs, were degraded prior to amplification by treatment of the DNA templates with plasmid-safe adenosine triphosphate (ATP)-dependent deoxyribonuclease DNase (Epicentre Technologies). In brief, HBV-infected cells were lysed for 4 hr at 65°C in lysis buffer (50 mM Tris–HCl, pH 8.0, 50 mM EDTA, 100 mM NaCl, 1% SDS) supplemented with proteinase K (200 µg/ml) and followed by phenol-chloroform extraction. A total

of 250 ng of the extracted DNA was digested with 5–10 units plasmid-safe DNase in a 50 μl volume for 8 hr at 37°C followed by DNase inactivation at 70°C for 30 min. 2 μl of the 50 μl reaction was then added to 20 μl of a real-time PCR reaction. Amplification of 735 bp cccDNA product was conducted by denaturation at 95°C for 5 min, followed by 45 cycles of denaturation at 95°C for 30 s, 62°C annealing for 25 s, and 72°C elongation for 45 s. HBV cccDNA copy numbers were calculated with a standard curve from plasmid with known nucleic acid quantities. The detection limit for cccDNA is ~10 copies cccDNA per reaction (equivalent to 10 ng of total cell lysate DNA). Real-time PCR was performed with SYBR Premix Ex Taq kit on an ABI Fast 7500 real-time system instrument.

## PTH cDNA library construction, deep sequencing of *Tupaia* transcriptome, and bioinformatics analysis of Illumina deep sequencing-determined transcriptome

Primary *Tupaia* hepatocytes were isolated as described above. PTH mRNA was purified from 10 μg of total RNA using Oligo-dT magnetic beads. The mRNA was fragmented into small pieces by incubation with divalent cations at 94°C for exactly 5 min. The first strand cDNA was synthesized using random primers and SuperScript II reverse transcriptase (Invitrogen) with fragmented mRNAs. RNA template was then removed by RNase H, and double-stranded cDNA was prepared with DNA polymerase I. cDNA with blunt ends was created by T4 DNA polymerase and Klenow DNA polymerase and an 'A' base was subsequently added to the 3′ end of the blunt phosphorylated DNA fragments by Klenow fragment (3′ to 5′ exo minus). The cDNA was then ligated with adapters and then ran on a 2% agarose gel. The fragments with a size range from 200 ± 25 bp were purified, followed by amplification using the manufacturer's primers. The PCR products were then purified using QIAquick PCR purification Kit (Qiagen), quantified and diluted for cluster generation and deep sequencing. The 72-cycle pair-end sequencing was performed with Sequencing Kits (Version 5) on an Illumina Genome Analyzer IIx (Illumina, San Diego, United States). Illumina CASAVA pipeline v1.8.1 was used for sequence extraction and filtering.

## Bioinformatics analysis of Illumina deep sequencing-determined transcriptome of primary *Tupaia* hepatocytes

### De novo reconstruction of transcriptome from cDNA library deep sequencing data

Total 253,919,616-pair 72 nt sequences with 36.6G base from the sequencing results of the hepatocyte cDNA library described above were fed to Trinity (*Grabherr et al., 2011*) r20110519 using pair end RNA-seq protocol, with which 209,063 transcripts of average length of 1421 nt (minimum 300 nt, maximum 21,043 nt, and scaffold N50 of 3674 nt) were generated.

### Protein sequences identification and annotation

Following assembly, GENSCAN (*Burge and Karlin, 1997*) was used with default parameters to identify coding sequences and the encoded protein sequences of these transcripts. A total of 91,479 protein sequences were identified. Each chosen protein sequence was first annotated with its corresponding blastp (*Camacho et al., 2009*) matches from NCBI human protein sequences. Those not annotated in the first step were then submitted for similar annotation process with UniprotKB human proteome and NCBI nonredundant protein sequence database. Protein sequences that were not annotated by previous steps were submitted for annotation with their corresponding transcripts. Protein sequences with their corresponding transcripts that can be annotated by the blastx (*Camacho et al., 2009*) hits of NCBI human protein sequences, UniProtKB human proteome or NCBI nonredundant protein sequence database were annotated with these hits from transcripts.

### Generation of *Tupaia* hepatocyte protein sequences database

All identified protein sequences were included in the hepatocyte protein sequences database. The protein sequences were labeled with corresponding functional annotation results. Any identified protein sequences that were not successfully annotated are labeled with 'Uncharacterized protein'. Total 50,951 annotated and 40,528 uncharacterized protein sequences generated from the *Tupaia* hepatocytes transcriptome were combined into the database. A numeric ID (gi) was

generated for each protein sequence in the database. The corresponding cDNA sequences were deposited to NCBI Transcriptome Shotgun Assembly (TSA) database of GenBank with accession number from JU120276 to JU170736 after removal of cDNAs shorter than 200 bp and a vector sequence.

### Analysis of protein classes in *Tupaia* hepatocytes

PANTHER database (*Mi et al., 2005*) was used to determine the protein class distribution of annotated transcripts and protein sequences in primary *Tupaia* hepatocytes (PTHs) generated in this study and transcriptome from primary human hepatocytes (PHHs) reported by *Hart et al. (2010)*.

## Photo-cross-linking with peptide ligand and tandem purification of the target molecule(s)

L-photo-leucine-bearing wild-type bait peptide ($WT_b$) or control bait peptide ($N9K_b$) was dissolved in DMSO in dark. L-photo-leucine contains a photoactivatable diazirine ring, irradiation of UV light at 365 nm induces a loss of nitrogen of the diazirine ring, and yields a reactive carbene group with short half-life for covalent cross-linking at nearly zero distance. For tandem purification, $WT_b$ or $N9K_b$ at indicated concentrations was applied to ~$1 \times 10^7$ hepatocytes plated on collagen-coated dishes. Cells were cross-linked by UV irradiation and then washed to remove residual free peptides and subsequently lysed with 1 ml radioimmunoprecipitation assay (RIPA, pH 7.4) buffer containing 20 mM Tris, 150 mM NaCl, 0.1% SDS; 0.5% sodium deoxycholate, 1% NP40, and 1× protease inhibitor cocktail (Roche). The cell lysates were precipitated with 100 µl streptavidin T1 magnetic beads, eluted with 50 µl nonreducing SDS-PAGE loading buffer and then diluted with cold RIPA buffer to a final volume of 1 ml and was precipitated with 100 µl ($1 \times 10^8$) 2D3-conjugated M-270 dynabeads, then eluted with 100 µl nonreducing loading buffer. The elute, with or without PNGase F treatment, was diluted to 1 ml with RIPA buffer and was precipitated again with 100 µl streptavidin T1 magnetic beads, followed by extensive washing, and finally eluted by boiling 5 min with 20 µl SDS-PAGE loading buffer. The samples were then analyzed with 12% SDS–PAGE and silver staining. For photo-cross-linking analysis of primary cells, cell lines, or NTCP- or control plasmid-transfected Huh-7 or 293T cells, $WT_b$ or $N9K_b$ bait peptide in the presence or absence of competing peptide was applied to $2 \times 10^6$ cells, photo-cross-linking was conducted similarly as described above. The cross-linked samples were precipitated with streptavidin T1 magnetic beads and separated by SDS-PAGE, and followed by Western blotting with mAb 2D3 (recognizing bait peptides) or mAb 1D4 against C-terminal tag C9.

## LC-MS/MS and data analysis

Silver stained gel bands were cut, followed by in-gel reduction, alkylation, and trypsin digestion as previously described (*Shevchenko et al., 2006*). Digested peptide mixtures containing 0.1% formic acid were loaded onto a 4 cm, 75-µm inner diameter fused silica capillary column packed with 10-µm YMC C18 material (YMC, Kyoto, Japan). After desalting, samples were separated with a Waters nano ACQUITY UltraPerformance LC (Waters, United States) and eluted to a LTQ-Orbitrap Velos mass spectrometer (Thermo Fisher Scientific, United States). The UPLC separation gradient included a 30-min gradient from 0% to 30% acetonitrile, followed by a 10-min gradient to 80% acetonitrile, then 10 min of 80% acetonitrile and back to 0% acetonitrile within 5 min. The mass spectrometer was operated in the data-dependent mode. Survey MS scans were acquired in the orbitrap with the resolution set to a value of 60,000. Each survey scan (300–2000 m/z) was followed by four data-dependent CID tandem mass (MS/MS) scans at 35% normalized collision energy and four data-dependent HCD tandem mass (MS/MS) scans at 40% normalized collision energy with 15,000 resolution in orbitrap. AGC target values were 500,000 for the survey scan, 10,000 for the ion trap MS/MS scan, and 50,000 for the orbitrap MS/MS scan. Target ions already selected for MS/MS were dynamically excluded for 30 s. Tandem mass spectra were searched against the Illumina deep sequencing-determined *Tupaia* hepatocyte protein database that was concatenated with reversed sequences to estimate false positives and was supplemented with the sequence of bait peptide under the Linux operating system using the ProLuCID (*Xu et al., 2006*) protein database search algorithm with peptide mass tolerance of ±100 ppm, fragment ion mass

tolerance of ±400 ppm, half tryptic specificity, and a static modification of 57.0215 on Cys due to carboxyamidomethylation. ProLuCID search results were then filtered with DTA Select 2.0 (*Tabb et al., 2002*) using a cutoff of 1% for peptide false identification rate (–fp 0.01). Peptides with DeltaMass > 10 ppm (–DM 10) were rejected; the minimum number of peptides to identify a protein was set to 1 (–p 1).

## Immunofluorescence microscopy and FACS analysis of NTCP binding with pre-S1 peptides

Plasmid encoding human, or *Tupaia* NTCP, with or without a tag, or a control plasmid was transfected into cells. The cells were washed and blocked with 3% BSA 24–36 hr after transfection and then stained with biotin-labeled peptides at 4°C about 1–2 hr, followed by fixation with 4% paraformaldehyde (PFA) for 10 min. Cells were then stained with PE-labeled Streptavidin (eBioscience). In some cases, Myr-59 peptide containing the first 59 residues of pre-S1 domain with an N-terminal myristoylation modification and labeled with FITC (FITC-pre-S1) was applied to cells directly. The cell images were captured with a Nikon Eclipse T*i* Fluorescence Microscope or a Zeiss LSM 510 Meta Confocal Microscope. For FACS analysis, the FITC-pre-S1 stained cells without fixation were detached with 0.5 mM EDTA/PBS, washed and resuspended in PBS, and analyzed with a FACS LSRII instrument (BD).

## Viral infections on cells treated with siRNAs

### For gene knockdown experiment on PTH cells

Four siRNAs (tsNTCP-si1: 5′-CUAUGUAGGCAUUGUGAUAdTdT-3′, tsNTCP-si2: 5′-GUGUUAUCCU GGUGGUUAUdTdT-3′, tsNTCP-si3: 5′-GGACAUGAAUCUCAGCAUUdTdT-3′, tsNTCP-si4: 5′-GGGCAAGAGCAUCAUGUUUdTdT-3′) specifically targeting *Tupaia slc10a1* were designed through siDESIGN Center (www.thermo.com/sidesign). The specificities of these siRNAs were examined by searching against NCBI cDNA databases and the in-house *Tupaia* hepatocyte transcriptome to ensure they are free of off-target. siRNA with sequence Ctrl-si: 5′-UUCUCCGAACGUGUCACGUdTdT-3′ that is a scramble sequence with no known mammalian target sequence was used as a negative control. 20 nM siRNA were transfected into PTHs 24 h after initial seeding with lipofectamine 2000 (Invitrogen). tsNTCP mRNA level in the siRNA-transfected cells was quantified by real-time RT-PCR 3 days after siRNA transfection. Transfected PTH cells were infected on day 4 after siRNA transfection with HBV, HDV, AAV8-HBV (carrying 1.05 copies of HBV genome), or Lenti-VSV-G-Luc viruses. The secreted viral antigen HBsAg or HBeAg from HBV- and AAV8-HBV-infected cells were examined as indicated. For Lenti-VSV-G-Luc virus-infected cells, luciferase activity was determined on 6 days post-infection.

### For gene knockdown experiment on HepaRG cells

*SLC10A1* from HepaRG cells was cloned, and the sequence was deposited to Genebank (accession number: JQ814895). Differentiated HepaRG cells in 48-well plate were transfected using RNAiMax (Invitrogen) with 20 nM of a siRNA pool (Qiagen) containing four specific siRNAs targeting different regions of human *SLC10A1* (5′-GGAUCGUCCUCAAAUCCAAdTdT-3′, 5′-GGAGUCAGCCGGAGAACAAdTd T-3′, 5′-GGACAAGGUGCCCUAUAAAdTdT-3′, 5′-GGUGCUAUGAGAAAUUCAAdTdT-3′) or a negative control siRNA (Ctrl-si: 5′-UUCUCCGAACGUGUCACGUdTdT-3′) as indicated. Gene knockdown efficiency was examined 4 days after transfection. Cells were then inoculated for 16 hr with HBV in the presence of 3.6% PEG8000. The secreted HBeAg and the 3.5 kb HBV RNA were determined on indicated days after infection.

### For gene knockdown experiment on PHH cells

Frozen PHH cells were thawed and subsequently transfected as that of PTH with human *SLC10A1*-specific siRNA 5′-CACAAGUGCUGUAGAAUUAdTdT-3′ (siR405) or a siRNA pool containing four *SLC10A1* specific siRNAs from QIAGEN (5′-GGAUCGUCCUCAAAUCCAAdTdT-3′, 5′-GGAGUCAGCCGGAGAACAAdTdT-3′, 5′-GGACAAGGUGCCCUAUAAAdTdT-3′, 5′-GGUGCUA UGAGAAAUUCAAdTdT-3′). Control siRNA 5′-UUCUCCGAACGUGUCACGUdTdT-3′ (Ctrl-si) has a scramble sequence with no known mammalian target sequence. Total 20 nM siRNA was transfected into ~1.4 × 10^5 PHH cells per well in 48-well plate 24 h after initial cell seeding; the cells were then inoculated with HBV 72 hr after transfection. hNTCP mRNA level in the

siRNA-transfected cells was quantified by qRT-PCR 3 days after siRNA transfection. The secreted antigens HBsAg and HBeAg, and the intracellular HBV RNAs were determined on indicated days after infection.

## HBV and HDV infection on receptor complemented cells

HepG2 or Huh-7 cells were transfected with a plasmid expressing human, treeshrew, monkey NTCP, or an NTCP variant, or a vector control. Stable cell line expressing hNTCP was established by transfection of HepG2 cells with a plasmid encoding hNTCP (hNTCP/pcDNA3.1) followed by selection with 500 µg/ml G418 and maintained in DMEM supplemented with 10% FBS, 500 µg/ml G418, 100 U/ml penicillin, and 100 µg/ml streptomycin. The transfected cells, or HepG2-hNTCP stable cells, were cultured in PMM 24 hr before infection.

### For HDV infection

With $5 \times 10^7$ genome equivalent copies of HDV, $1 \times 10^5$ cells were incubated, or otherwise indicated, for 24 hr in the presence or absence of entry inhibitors. HDV inoculation was conducted in the presence of 4% PEG8000. PMM was replenished every 2 days. On indicated days post-infection, cells were treated with 100% methanol for 10 min, followed by incubating with 5 µg/ml FITC-labeled mAb 4G5 for 1 hr at RT to stain HDV delta antigen. The nuclei were stained with 4′-6-diamidino-2-phenylindole (DAPI) before analyzing. HDV viral RNA copies in cell lysates were quantified by qPCR.

### For HBV infection

With ~$1 \times 10^7$ genome equivalent copies of HBV, $1 \times 10^5$ cells were inoculated, or indicated otherwise, in the presence of ~4% PEG8000 as reported for primary human hepatocyte and HepaRG cell (*Gripon et al., 1993*; *Gripon et al., 2002*; *Schulze et al., 2007*). The cells were maintained subsequently in PMM and the medium was changed every 2–3 days. For immunofluorescence microscopy analysis, HBV-infected cells, with or without replating on glass coverslips for imaging, were fixed with 4% Paraformaldehyde (PFA) and permeabilized with 0.5% TritionX-100, and then stained either with 10 µg/ml 17B9 against HBsAg followed by FITC-labeled goat anti-mouse IgG, or with 5 µg/ml 1C10 against HBcAg followed by Qdot 655 VIVID donkey anti-mouse IgG. 1 µg/ml of DAPI was added to stain the nucleus before analyzing. The cell images were captured with a Nikon Eclipse Ti Fluorescence Microscope or a Zeiss LSM 510 Meta Confocal Microscope. Secreted viral antigens and intracellular viral replication intermediates cccDNA and/or RNAs were examined on indicated days after infection.

## Levels of total and surface protein expression of human and monkey NTCP variants

For cell surface expression, the transfected cells expressing NTCPs or mutants were surface-biotinylated with sulfo-NHS-LC-biotin (Pierce) following the manufacturer's instruction manual. The biotinylated cells were then lysed in 600 µl of 1× RIPA buffer supplemented with 1× protease inhibitor cocktail (Roche) and then treated with PNGaes F. Total cellular protein in the lysate was determined by using Bio-Rad DC Protein assay. About 300 µg Streptavidin T1 Dynabeads (Invitrogen) were then used to pull down surface-biotinylated proteins in the supernatants containing ~160 µg of total cellular protein each sample. After extensive washing with 1× RIPA buffer, bound proteins were eluted, and separated by SDS-PAGE and subsequently examined with anti-C9 mAb 1D4 by Western blotting. For total NTCP expression, the same transfected cells expressing wild-type or mutant NTCPs were lysed with 1× RIPA buffer and treated with PNGase F. Each sample containing same amount of total cellular protein (~8 µg) were loaded for SDS-PAGE followed by Western blotting analysis with mAb 1D4 that recognizes the C9 tag fused at the C-terminus of NTCPs.

## Data processing and analysis

All experiments were repeated 2–6 times with duplicate or triplicate samples for each condition unless indicated otherwise. A representative result of multiple independent experiments is present (n = 2–6) in each figure. Error bars shown in all figures represent standard deviation of the mean (n = 2–4). Dotted lines show detection limit except otherwise specified.

## Acknowledgements

We thank Dr. She Chen and Lin Li at National Institute of Biological Sciences (NIBS) proteomic center for the mass spectrometry analysis, Dr. Cheng Zhan at imaging center, and Zhihua Qiu and Ping Qu at animal facility of NIBS for their excellent technical assistances. We thank Dr. Michael R. Farzan for helpful discussions and Drs. Xiaodong Wang and Hyeryun Choe for critical comments on the article. We are grateful for Drs. Lin Jiang, Zhenglun Liang, and Fengming Lu for providing us with valuable reagents.

## Additional information

### Funding

| Funder | Grant reference number | Author |
| --- | --- | --- |
| The Ministry of Science and Technology of the People's Republic of China | 2010CB530101 | Huan Yan, Guocai Zhong, Pan Chen, Yinyan Sun, Wenhui Li |
| Science and Technology Bureau of Beijing Municipal Government | | Huan Yan, Guocai Zhong, Guangwei Xu, Wenhui He, Zhiyi Jing, Zhenchao Gao, Yi Huang, Yonghe Qi, Bo Peng, Haimin Wang, Liran Fu, Mei Song, Pan Chen, Wenqing Gao, Bijie Ren, Yinyan Sun, Tao Cai, Xiaofeng Feng, Jianhua Sui, Wenhui Li |
| The Ministry of Science and Technology of the People's Republic of China | 2011CB812501 | Huan Yan, Guocai Zhong, Guangwei Xu, Wenhui He, Zhiyi Jing, Zhenchao Gao, Yi Huang, Yonghe Qi, Bo Peng, Haimin Wang, Liran Fu, Mei Song, Pan Chen, Wenqing Gao, Bijie Ren, Yinyan Sun, Tao Cai, Xiaofeng Feng, Jianhua Sui, Wenhui Li |

The funders had no role in study design, data collection and interpretation, or the decision to submit the work for publication.

### Author contributions

HY, Conception and design, Acquisition of data, Analysis and interpretation of data; GZ, Conception and design, Acquisition of data, Analysis and interpretation of data; GX, Acquisition of data, Analysis and interpretation of data; WH, Acquisition of data, Analysis and interpretation of data; ZJ, Acquisition of data, Analysis and interpretation of data; ZG, Acquisition of data, Analysis and interpretation of data; YH, Acquisition of data, Analysis and interpretation of data; YQ, Acquisition of data; BP, Acquisition of data; HW, Acquisition of data; LF, Acquisition of data; MS, Acquisition of data; PC, Acquisition of data; WG, Acquisition of data; BR, Acquisition of data; YS, Acquisition of data; TC, Acquisition of data, Analysis and interpretation of data; XF, Acquisition of data, Analysis and interpretation of data; JS, Conception and design, Analysis and interpretation of data, Drafting or revising the article; WL, Conception and design, Analysis and interpretation of data, Drafting or revising the article

### Ethics

Human subjects: This study was approved by the Institutional Review Board (IRB) of National Institute of Biological Sciences (IRBS030901) and consent was obtained; see the Materials and Methods section for details. The Helsinki guidelines were followed.
Animal experimentation: The institutional animal care and use committee (IACUC) of the National Institute of Biological Sciences and the approved animal protocol is 09001T. The institutional guidelines for the care and use of laboratory animals were followed.

## Additional files

### Major datasets

The following datasets were generated

| Author(s) | Year | Dataset title | Dataset ID and/or URL | Database, license, and accessibility information |
|-----------|------|---------------|------------------------|-------------------------------------------------|
| Jing Z, Gao Z, Xu G, He W, Cai T, Li W | 2012 | Tree shrew (*Tupaia belangeri chinensis*) hepatocyte transcriptome assembly project | JU120276-JU170736; http://www.ncbi.nlm.nih.gov/genbank/ | In the public domain at the Transcriptome Shotgun Assembly Database |

**Reporting standards:** We followed the reporting standards for the Transcriptome Shotgun Assembly Sequence Database of NCBI, which are: Submitted sequences must be assembled from data experimentally determined by the submitter. Screened for vector contamination and any vector/linker sequence removed. This includes the removal of NextGen sequencing primers. Sequences cannot be less than 200 bp. Sequences should have no more than 10% n's or greater than 14 n's in a row. If the submission is a single-step, unannotated assembly and the output is a BAM file(s) these should be submitted as a TSA project to SRA.

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
