## [Decision Letter]

Thank you for choosing to send your work entitled “Sodium taurocholate cotransporting polypeptide is a functional receptor for human hepatitis B and D virus” for consideration at *eLife*. Your article has been evaluated by a Senior Editor and 3 reviewers, one of whom is a member of *eLife's* Board of Reviewing Editors.

The Reviewing Editor and the other reviewers discussed their comments before we reached this decision, and the Reviewing Editor has assembled the following comments based on the reviewers' reports. Our goal is to provide the essential revision requirements as a single set of instructions, so that you have a clear view of the revisions that are necessary for us to publish your work.

This manuscript convincingly identifies NTCP as a receptor for HBV and HDV. The evidence in support of this conclusion includes: a) RNAi of NTCP in the virus-susceptible PTH and PHH cells reduced the virus infection; b) expression of NTCP in HBV non-permissive cell lines render these cells sensitive to HBV and HDV infections; c) swapping experiments showed that grafting of just 9 amino acids of human NTCP into monkey NTCP converts the monkey protein into a functional HBV/HDV receptor; d) The permissiveness of various cell lines to HBV/HDV infection correlates very well with the expression levels of NTCP; e) cross-linking experiments suggest that NTCP binds to the pre-S1 domain of the L envelope protein. The discovery of NTCP as the long-sought receptor for HBV and HDV is a major advance in the field and will have a large impact on both basic and clinical research in viral hepatitis.

This paper may be improved by addressing the following points:

1) It is not clear whether the viruses released from Huh7-NTCP or HepG2-NTCP could infect primary human hepatocytes (PHH). If so, what's the infectious titer?

2) Only 5–10% of stable HepG2-hNTCP cells were infected with HBV. How does this compare to primary human hepatocytes? An explanation for this low infection efficiency should be provided.

---

## [Author Response]

*1) It is not clear whether the viruses released from Huh7-NTCP or HepG2-NTCP could infect primary human hepatocytes (PHH). If so, what's the infectious titer*?

To address this question, we performed new experiments. We found inoculation of primary human hepatocytes with the culture medium collected from HBV infected NTCP-expressing HepG2 cells only led to very limited infection on PHHs. This very low level of 2^nd^ round infection was likely due to limited viral particles released from infected HepG2-NTCP cells. Indeed, only small amount of viral DNA was detected in the culture medium from the infected cells up to 13 days post primary infection. This is similar to HBV infection of PHHs. It has been known that HBV virion release from infected primary hepatocytes in culture is inefficient [1,2], only a level of less than 1% input virions could be detected in the culture medium at the maximum of HBV replication on PHH [2]. In our own experiments, the viral DNA level in culture medium is comparable between infected PHH and HepG2-NTCP cells: in both cultures, a level equivalent to ∼1% of input viral genome copies was detected. We included these data in new Figure 6—figure supplement 3. It is tempting to speculate that some host factors that are lacking in cell cultures *in vitro* might be needed for efficient viral particles formation or releasing; or some cellular factors in these cultures may hinder these processes during infection. Further studies are required to understand how the interactions between host cells and HBV viruses control the multiple steps of viral propagation and spreading after primary viral entry. The description for the new Figure 6—figure supplement 3 was added in the Results section.

*2) Only 5–10% of stable HepG2-hNTCP cells were infected with HBV. How does this compare to primary human hepatocytes? An explanation for this low infection efficiency should be provided*.

To compare the efficiency of HBV infection on HepG2-hNTCP cells and PHHs, we performed side-by-side experimental tests in which HepG2-hNTCP and PHHs from two independent donors were inoculated with same doses of HBV viruses, respectively. The results are shown in new Figure 6—figure supplement 2, with text added in the Results section. By staining intracellular HBV core antigen (HBcAg) on day 8 post infection (dpi 8), we estimated that about 10% of HepG2-hNTCP cells were infected at multiplicity of genome equivalents (mge) 100. For the two PHHs, the infection efficiency was up to ∼5% and ∼10%, respectively. HepG2-NTCP cells but not PHHs propagate in cultures, thus the actual infection efficiency of HepG2-hNTCP cells may be underestimated by the observed infection using HBcAg staining at the endpoint of infection. Nevertheless, the infection efficiency in HepG2-NTCP cells is comparable to PHHs of good quality at the same infection dose under the conditions tested. It has been reported that up to 5% of PHHs could be infected *in vitro* at ∼80 mge [3], more efficient infection on primary hepatocytes required HBV of thousands mge [3,4]. This is in contrast to the *in vivo* infection of HBV in chimpanzees, for which much lower HBV mge is sufficient [5]. It is possible that the microenvironment and architecture of hepatocytes in liver or other host factors, such as soluble ones in blood or other co-receptor/entry components that are lacking or lost in cultures, may also contribute to the efficiency of HBV infection.

**References:**

1 Gripon, P. et al. Hepatitis B virus infection of adult human hepatocytes cultured in the presence of dimethyl sulfoxide. J Virol 62, 4136–4143 (1988).

2 Boehm, ST., Thasler, WE., Weiss,TS., Jilg,W. in Models of Viral Hepatitis Vol. 25 (ed Roggendorf M, Von Weizsacker F) 119–134 (Karger, 2005).

3 Schulze, A., Mills, K., Weiss, T. S. & Urban, S. Hepatocyte polarization is essential for the productive entry of the hepatitis B virus. Hepatology 55, 373–383, (2012).

4 Glebe, D. et al. Pre-s1 antigen-dependent infection of Tupaia hepatocyte cultures with human hepatitis B virus. J Virol 77, 9511–9521 (2003).

5 Asabe, S. et al. The size of the viral inoculum contributes to the outcome of hepatitis B virus infection. J Virol 83, 9652–9662 (2009).